# KALE: Enhancing Knowledge Manipulation in Large Language Models via Knowledge-aware Learning

## Abstract

Despite the impressive performance of large language models (LLMs) pretrained on vast knowledge corpora, advancing their knowledge manipulation performance—the ability to effectively **recall, reason, and transfer relevant knowledge**—still remains challenging. Existing methods mainly leverage supervised fine-tuning (SFT) to enable LLMs to recall task-relevant knowledge by continuing the training process on labeled datasets. However, we observe that LLMs fine-tuned via SFT still occasionally exhibit the *known&incorrect* phenomenon, where LLMs explicitly possess the relevant knowledge of a given question but cannot effectively manipulate it to answer correctly. To address this challenge, we propose KALE—a novel post-training framework that leverages knowledge graphs (KGs) to generate high-quality relevant rationales and enhance the knowledge manipulation ability via **K**nowledge-**A**ware **LE**arning. Specifically, KALE **first** proposes a **K**nowledge-**I**nduced (KI) data synthesis method to generate high-quality data rationales, i.e., a textual reasoning process from each question to the correct answer through external KGs. **Then** KALE proposes a **K**nowledge-**A**ware (KA) fine-tuning paradigm to enhance the knowledge manipulation ability of LLMs. Extensive experiments on **eight** popular benchmarks across **six** different LLM backbones demonstrate the effectiveness of KALE, leading to an accuracy improvement of up to 11.72% and an average of 4.18%.

## 1 Introduction

Standing out as versatile tools with vast knowledge repositories, large language models (LLMs), such as GPT-4.5 (OpenAI, 2024), Deepseek R1 (Team, 2024b), LlaMA-3 (Touvron et al., 2023), and Qwen2.5 (Team, 2024d), demonstrate remarkable power and versatility across a wide range of domains (Zhao et al., 2021; El-Kassas et al., 2021). However, the most capable LLMs also produce errors, even when the knowledge is explicitly encoded within LLMs, indicating struggles for existing LLMs to flexibly manipulate task-relevant knowledge during inference (Allen-Zhu & Li, 2024; 2025).

Recently, extensive research efforts have been devoted to boosting LLM knowledge manipulation performance for downstream tasks. One promising post-training paradigm, Supervised Fine-Tuning (SFT), has emerged as a new trend, demonstrating superior performance in enhancing the ability of LLMs on certain downstream tasks (Wei et al., 2022a). The key idea of SFT is to adapt pre-trained LLMs to specific tasks by conducting the post-training process on labeled datasets, which refines their parameters to focus on task-relevant features (Zhang et al., 2023). Several endeavors also explored variations of SFT methods. Dual-stage Mixed fine-Tuning (DMT) (Dong et al., 2023) expands SFT datasets to achieve a balance between the general and specialized ability. KG-SFT (Chen et al., 2025) utilizes knowledge graphs (KGs) to filter SFT data to enhance LLMs' ability on knowledge-intensive tasks. Extensive studies further demonstrate both the effectiveness (Dong et al., 2023) and versatility (Xie et al., 2024) of SFT methods.

Albeit with multiple benefits of SFT methods, LLMs fine-tuned via SFT still exhibit the *known&incorrect* phenomenon—**LLMs possess relevant knowledge but cannot manipulate it to correctly answer questions.** This phenomenon mainly stems from two limitations in the SFT process: **(i) the lack of high-quality textual reasoning data from question to answer.** For certain

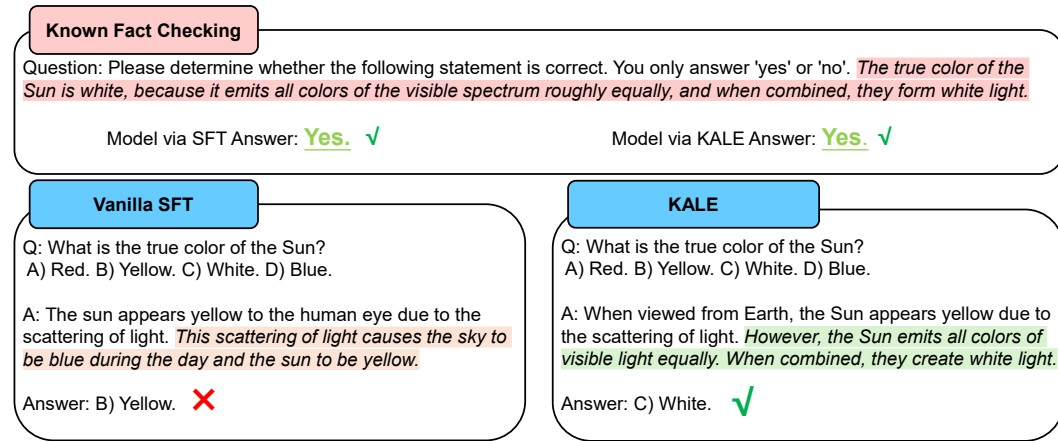

Figure 1: While both post-trained LLMs know relevant knowledge, the LLM via SFT still cannot recall the knowledge to answer. In contrast, KALE effectively recalls the knowledge and answers correctly. We use Mistral 7B (Jiang et al., 2023a) as an example, and more cases are in Appendix B.

domains, off-the-shelf reasoning data is scarce, and creating such data needs substantial human effort, which poses a significant barrier to broader applications of LLMs in downstream domains (Li et al., 2024) and **(ii) the insufficient ability to recall task-relevant knowledge**. SFT methods fine-tune LLMs using labeled datasets, where LLMs learn specific patterns through explicit input-output pairs. However, LLMs often overly rely on explicit input-output mappings, which restrict their ability to dynamically retrieve task-relevant knowledge (Luo et al., 2024). As shown in the left part of Figure 1, despite being explicitly known that the true color of the Sun is white, LLM after SFT still cannot recall this knowledge to provide a correct answer. **As a result, even after a sufficient SFT process, LLMs still struggle to effectively manipulate task-relevant knowledge to answer correctly for downstream tasks** (Allen-Zhu & Li, 2025).

To address these challenges, we propose a novel post-training framework, namely **K**nowledge-**A**ware **LE**arning (KALE) to boost LLM's knowledge manipulation abilities. KALE consists of two components: (i) knowledge-induced data synthesis (**KI**) to generate high-quality rationale data and (ii) knowledge-aware fine-tuning (**KA**) to enable LLMs to manipulate task-relevant knowledge. Specifically, for a given Q&A pair, KALE **first** identifies named entities within the pair and extracts reasoning paths from question to answer via the proposed multi-path A* algorithm through an external KG. **Then**, KALE combines the pair and reasoning paths as input for the LLM to generate rationales underlying the pair. **Finally**, rather than learning specific patterns through explicit supervised input-output pairs, KALE minimizes the KL divergence (Kullback & Leibler, 1951) between LLM distributions with and without rationales. By doing so, KALE does not require the outputs without rationales to exactly match those produced with rationales. Instead, KALE encourages the two distributions to be more aligned, which allows LLMs to more flexibly recall task-relevant knowledge when rationales are absent during inference.

We summarize our major contributions as follows:

(i) **An efficient high-quality SFT data generation method.** We propose an autonomous high-quality SFT data synthesis method to generate text reasoning rationales for each Q&A pair to improve the comprehensive ability of LLMs in understanding the underlying logic behind the Q&A questions.

(ii) **A flexible knowledge manipulation fine-tuning paradigm.** We propose a knowledge-aware fine-tuning paradigm to encourage LLMs to recall relevant knowledge when answering questions by aligning distributions of LLMs with and without input rationales.

(iii) **Significant Improvement and Versatility.** We conduct extensive experiments on **eight** different popular downstream benchmarks on **six** different LLM backbones to demonstrate the effectiveness of our KALE with a maximum accuracy improvement of 11.72%.

## 2 RELATED WORK

### 2.1 TEXT DATA AUGMENTATION METHODS

With the advent of LLMs, data augmentation has undergone a significant transformation (Ding et al., 2024). LLMs have shown remarkable abilities in generating high-quality text, which provides significant advantages in data augmentation tasks (Deng et al., 2023; Fang et al., 2023). AugGPT (Dai et al., 2023) leverages the generative power of LLMs to rephrase questions in SFT data. GPT3Mix (Yoo et al., 2021) extends the data augmentation abilities of LLMs by using few-shot prompting to generate questions semantically similar to the SFT data. StaR (Zelikman et al., 2024) utilizes a self-taught mechanism to let LLMs provide internal thoughts. While existing data augmentation methods primarily focus on expanding the data quantity but lack the multi-hop logic rationale, **our KALE can effectively generate textual rationales underlying the Q&A pair.**

### 2.2 KNOWLEDGE GRAPH RETRIEVAL GENERATION METHODS

Knowledge graphs (KGs) offer a complementary way to the unstructured, text-based knowledge encoded in LLMs (Pan et al., 2024). Recent research has explored the integration of KGs to enhance the Q&A and reasoning abilities of LLMs. Think-on-Graph (ToG) (Sun et al.) employs an iterative beam search over a KG to guide the reasoning process of LLMs. KGR (Guan et al., 2024) retrofits LLM responses with factual statements from KGs. KAPING (Baek et al., 2023) enhances zero-shot Q&A by appending retrieved facts to LLMs. StructGPT (Jiang et al., 2023b) employs an iterative reading-then-reasoning framework to reason over structured data. GraphRAG (Edge et al., 2024) integrates KG traversal to retrieve structured relationships from graph-indexed data. Existing retrieval-based methods require additional retrieval from a knowledge base during inference, resulting in extra time overhead. **Our KALE, once trained, does not necessitate any additional time consumption during inference (We provide an average testing time per sample of KALE in Appendix G).**

### 2.3 SFT VARIANTS METHODS

With the rise of LLMs, there is a growing emphasis on using SFT to align LLMs with human intentions to downstream tasks (Ouyang et al., 2022). Many innovative fine-tuning strategies have been proposed to enhance the performance and adaptability of LLMs. Dual-stage Mixed fine-Tuning (DMT) (Dong et al., 2023) proposes to improve the general ability of LLMs, making them more adept at handling diverse tasks and domains. Self-Distillation Fine-Tuning (SDFT) (Yang et al., 2024) uses a distilled dataset generated by model itself during the fine-tuning to reduce the catastrophic forgetting (Kirkpatrick et al., 2017). KG-SFT (Chen et al., 2025) utilizes KGs to filter SFT data to enhance LLMs' ability on knowledge-intensive tasks. Existing SFT-based methods learn specific patterns through explicit input-output pairs, which restricts LLM's ability to dynamically retrieve task-relevant knowledge. **Our KALE enables more flexible manipulation of task-relevant knowledge.**

## 3 PRELIMINARIES

### 3.1 NOTATIONS

We denote $\mathbf{x}^{\text{ins}}$ as instructions for downstream tasks, $\mathbf{x}^{\text{que}}$ as queries, $\mathbf{x}^{\text{ans}}$ as answers and $\mathbf{x}^{\text{rats}}$ as rationales. We denote two types of input prompts for the LLMs as follows: one includes the rationale, represented as $(\mathbf{x}^{\text{ins}}, \mathbf{x}^{\text{que}}, \mathbf{x}^{\text{rats}})$, and the other excludes the rationale, as $(\mathbf{x}^{\text{ins}}, \mathbf{x}^{\text{que}})$. Let $\mathcal{E}_q = [\mathbf{e}_{q_1}, \mathbf{e}_{q_2}, \mathbf{e}_{q_3}, \ldots]$ denote the question entity list of $\mathbf{x}^{\text{que}}$, $\mathcal{E}_a = [\mathbf{e}_{a_1}, \mathbf{e}_{a_2}, \mathbf{e}_{a_3}, \ldots]$ denote the answer entity list of $\mathbf{x}^{\text{ans}}$, and $\mathcal{P} = [\mathbf{p}_1, \mathbf{p}_2, \mathbf{p}_3, \ldots]$ denote the reasoning path list connecting the question entity list to the answer entity list, where $e_{q_i}$ and $e_{a_i}$ denote the $i$-th entity of $\mathbf{x}^{\text{que}}$ and $\mathbf{x}^{\text{ans}}$ and $\mathbf{p}_i$ denotes the $i$-th path of the reasoning path list $\mathcal{P}$. Let $g(\mathbf{e})$, $h(\mathbf{e})$, and $f(\mathbf{e})$ be the current accumulated cost, heuristic estimated cost, and total estimated cost for a given entity $\mathbf{e}$, respectively.

### 3.2 A* ALGORITHM

A* algorithm (Hart et al., 1968) is an extension of the Bellman-Ford algorithm (Bellman, 1958; Ford, 1956; Moore, 1959). Unlike the Bellman-Ford algorithm that propagates through nodes uniformly,

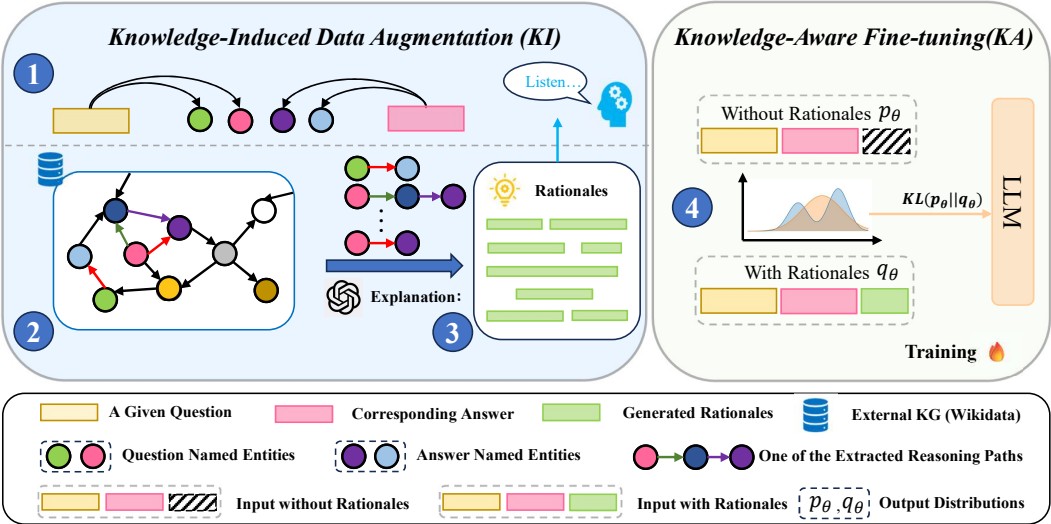

Figure 2: An overview of KALE. For a given Q&A pair in the training set, the workflow of KALE is as follows. **(1)** Perform named entity recognition to extract potential question and answer entities. **(2)** Search the reasoning path via the proposed multi-path A* algorithm. **(3)** Combine the reasoning path and Q&A pair and generate the corresponding rationale via GPT-4o. **(4)** Align LLM's output distributions for cases with and without the rationale via knowledge-aware fine-tuning.

the A* algorithm prioritizes propagation with a proper heuristic function to reduce the search space:

$$s(\mathbf{e}) = d(\mathbf{e}_{start}, \mathbf{e}) \oplus h(\mathbf{e}, \mathbf{e}_{end}), \tag{1}$$

where $\oplus$ is a aggregation function, $d(\mathbf{e}_{start}, \mathbf{e})$ is the length of current shortest path from start entity $\mathbf{e}_{start}$ to $\mathbf{e}$, and $h(\mathbf{e}, \mathbf{e}_{end})$ is a heuristic function estimating the cost from $\mathbf{e}$ to target entity $\mathbf{e}_{end}$.

## 4 METHOD

### 4.1 KNOWLEDGE-INDUCED DATA SYNTHESIS

Answering a question may require the integration of multiple knowledge fragments. For instance, for the question *"What is the true color of the Sun?"* and answer: *"White"*, it involves multiple knowledge such as: **(i)** "The Sun emits all colors of visible spectrum," **(ii)** "The combination of all visible light produces white light," and **(iii)** "The intensity distribution of sunlight roughly exhibits balance when integrated." The fragmented nature of such knowledge in pre-training data creates challenges for LLMs to manipulate relevant knowledge during inference. In contrast, KGs provide a way to represent fragmented knowledge into structured and logical correlations. Specifically, it can be formalized into a reasoning path: "*the Sun–emits–>full spectrum light–integrates_into–>white light*," which corresponds to a series of interconnected triples within a KG, including [the Sun, emits, full-spectrum light] and [full-spectrum light, integrates into, white light]. Building upon the observation, we propose knowledge-induced data synthesis (KI) to generate rationales.[1]

Specifically, KALE will **first** perform named entity recognition separately on the question and the answer, resulting in the question entity list $\mathcal{E}_q = \{$the Sun$\}$ and the answer entity list $\mathcal{E}_a = \{$white$\}$. **Then**, KALE leverages these entities to search for reasoning paths in a KG. Conducting a full breadth-first search (BFS) from the question entities to the answer entities in a large KG (e.g., Wikidata[2]) is time-consuming. For instance, the extraction of reasoning paths from the AbsR's (Xiong et al., 2024)

---

[1]We only generate rationales for the training set. We use this query only to explain our workflows. In testing (as in Figure 1), we do not perform these steps. KALE does not introduce additional overhead in testing.

[2]In this paper, we use Wikidata by default as the external KG to extract all reasoning paths. To evaluate the robustness of KALE to different KGs, we report results using alternative KGs in Appendix M.

training set **requires over one week**. Therefore, we propose an efficient multi-path A* algorithm to extract reasoning paths. It requires **less than 4 hours** to extract all reasoning paths on the same set. Specifically, we adopt a small set of *anchor entities*. For a given entity pair $\mathbf{e}_q$ and $\mathbf{e}_a$ in $\mathcal{E}_q$ and $\mathcal{E}_a$, we select $k$ anchor entities by randomly sampling from the $m$-hop neighbors of the answer entity $\mathbf{e}_a$, thereby extracting a local subgraph around the answer entity. For each anchor, we conduct a limited 3-step BFS, i.e., a constrained BFS that explores up to three hops from the anchor entity to pre-compute partial distances, which serve as a lower bound for the remaining path cost in A*.

Formally, let $g(\mathbf{e})$ be the accumulated cost (the number of edges traversed) from start entity to current entity $\mathbf{e}$ and $h(\mathbf{e})$ be the heuristic function estimating the cost from $\mathbf{e}$ to the answer entity $\mathbf{e}_a$. We define the priority function as $f(\mathbf{e}) = g(\mathbf{e}) + h(\mathbf{e})$, where $f(\mathbf{e})$ is the priority value in A*. To ensure $h(\mathbf{e})$ does not overestimate the actual distance (preserving the admissibility condition of A*), we use the maximum of anchor-based lower bounds derived from the BFS. Specifically, let $\{\alpha_1, \alpha_2, \ldots, \alpha_k\}$ be $k$ anchor entities, we pre-compute $\mathrm{dist}(\alpha_i, \mathbf{e})$ up to depth $d$; if $\mathbf{e}$ is not reachable within $d$ steps, we set $\mathrm{dist}(\alpha_i, \mathbf{e}) = \infty$. Likewise, we compute $\mathrm{dist}(\alpha_i, \mathbf{e}_a)$ for each anchor. Then we let

$$h(\mathbf{e}) \;=\; \max_{1 \leq i \leq k} \Big[ \mathrm{dist}(\alpha_i, \mathbf{e}_a) \;-\; \mathrm{dist}(\alpha_i, \mathbf{e}) \Big]^+, \tag{2}$$

where $[x]^+ = \max(x, 0)$ ensures non-negative values. Intuitively, if $\mathbf{e}$ is already close to the answer entity compared with $\alpha_i$, this difference is a nontrivial lower bound; otherwise, it contributes zero and does not lead to overestimation (We prove the admissibility of our multi-path A* via the proposed heuristic function in Appendix C). This heuristic design is simple yet efficient for reasoning path retrieval in a large KG. We can also apply KG embedding-based methods (Rossi et al., 2021; Zhu et al., 2021; 2024) to incorporate semantic information from KG, and we leave it as future work.

To retrieve multiple reasoning paths, we extend the standard A* algorithm by incorporating a priority queue $\mathcal{Q}$, which stores multiple paths leading to the same entity. Each entry in $\mathcal{Q}$ is a tuple $(f(\mathbf{e}), g(\mathbf{e}), \mathbf{e}, p_{\mathbf{e}}^i)$, where $p_{\mathbf{e}}^i$ is the $i$-th path from the start entity $\mathbf{e}_q$ to the current entity $\mathbf{e}$. **Algorithm 1 in Appendix D provides the pseudo codes of the overall procedure.** After obtaining $\mathcal{P}$, we combine the Q&A pair and $\mathcal{P}$ as input for GPT-4o, prompting it to generate the rationale $\mathbf{x}^{\mathrm{rats}}$ underlying the Q&A pair (Appendix J.2 provides prompt details). For example, for the extracted reasoning path, "*the Sun–emits–>full spectrum light–integrates_into–>white light,*" the rationales are "*The Sun emits light that contains the entire visible spectrum. When these different colors of light are combined, they create white light.*" **These rationales offer high-quality textual reasoning data from question to answer, which enables better understanding of the underlying logic and correlations.** We include more examples of reasoning paths and rationales in Appendix K to provide a comprehensive understanding of KALE.

## 4.2 KNOWLEDGE-AWARE FINE-TUNING PARADIGM

When confronted with a given question, a typical human response process of answering often involves retrieving the related experiences and learned knowledge, reasoning based on this knowledge, and then providing a response (Buckner & Wheeler, 2001; Yadav et al., 2022). Motivated by this, we propose a simple yet effective learning paradigm called knowledge-aware fine-tuning, which encourages LLMs to recall relevant knowledge and reason over it before generating a response.

Formally, consider an LLM denoted by $\mathcal{M}$ with parameters $\theta$ and input $\mathbf{x}^{\mathrm{inp}} = (\mathbf{x}^{\mathrm{ins}}, \mathbf{x}^{\mathrm{que}}, \mathbf{x}^{\mathrm{ans}})$, where $\mathbf{x}^{\mathrm{ins}}$ denote instructions, $\mathbf{x}^{\mathrm{que}}$ and $\mathbf{x}^{\mathrm{ans}}$ denotes the Q&A pair. It constructs a conditional probability for the output $\mathbf{x}^{\mathrm{out}}$. We consider two probabilities, which differ in whether rationales are as input:

$$\mathcal{M}(\mathbf{x}^{\mathrm{inp}}, \mathbf{x}^{\mathrm{out}}, \theta) = -\sum_t \log p_\theta(\mathbf{x}_t^{\mathrm{out}} \mid \mathbf{x}^{\mathrm{inp}}, \mathbf{x}_{<t}^{\mathrm{out}}), \tag{3a}$$

$$\mathcal{M}(\mathbf{x}^{\mathrm{inp}}, \mathbf{x}^{\mathrm{rats}}, \mathbf{x}^{\mathrm{out}}, \theta) = -\sum_t \log q_\theta(\mathbf{x}_t^{\mathrm{out}} \mid \mathbf{x}^{\mathrm{inp}}, \mathbf{x}^{\mathrm{rats}}, \mathbf{x}_{<t}^{\mathrm{out}}). \tag{3b}$$

equation 3a represents the classical process of LLM generation, where a given instruction and query are provided as input, and the LLM produces an output. We aim for the LLM to manipulate learned knowledge and reason over it. As in equation 3b, we also use the generated rationales $\mathbf{x}^{\mathrm{rats}}$ as input to the LLM to enable better recalling knowledge fragments relevant to the question.

Therefore, we hope that the LLM can automatically complete rationales based on the instruction and query before generating a response, and we propose knowledge-aware fine-tuning to minimize the divergence between the two distributions in equation 3a and equation 3b as follows:

$$\mathcal{L}(\theta) = \mathbb{E}_{(\mathbf{x}^{\text{inp}}, \mathbf{x}^{\text{out}}, \mathbf{x}^{\text{rats}})} \left[ \text{KL} \left( p_\theta(\mathbf{x}_t^{\text{out}} \mid \mathbf{x}^{\text{inp}}, \mathbf{x}_{<t}^{\text{out}}) \| q_\theta(\mathbf{x}_t^{\text{out}} \mid \mathbf{x}^{\text{inp}}, \mathbf{x}^{\text{rats}}, \mathbf{x}_{<t}^{\text{out}}) \right) \right], \tag{4}$$

where $\text{KL}(\cdot \| \cdot)$ denotes the KL divergence. We initialize two LLMs: $p_\theta$ is updated during training, and $q_\theta$ is fixed and used solely as an alignment target. The latter represents the distribution with the input rationales that our KA aims to align. By minimizing the KL divergence in equation 4, KALE does not require outputs without rationales to exactly match those produced with rationales. Instead, it encourages the two distributions to align, which enables the LLM to flexibly retrieve task-relevant knowledge when rationales are absent during inference.

## 5 EXPERIMENTS

We aim to evaluate the effectiveness of KALE to enhance LLM's knowledge manipulation ability and the versatility of KALE. With this desiderata, we divide the experiments into **seven** parts:

- To demonstrate the superiority and generalizability, we conduct comparative experiments on **eight** different benchmarks across **six** different LLM backbones.
- To investigate the contribution of each component, we conduct the ablation study.
- To provide more insight, we conduct the case study on **known&incorrect phenomenon** and **ratios of augmented rationales**.
- To demonstrate the versatility, we evaluate our KALE on **knowledge-intensive domains** of six different languages in Appendix F.
- To analyze KALE's real-world deployability, we evaluate its **the inference time per sample and sensitivity to hyperparameters** in Appendix G and H.
- To provide a comprehensive understanding, we conduct an analysis of generated rationales:
  - (i) We employ **different external KGs** to generate reasoning paths in Appendix M.
  - (ii) We use **rationales by other LLMs** to demonstrate KALE's robustness in Appendix Q.
  - (iii) We prompt LLMs to generate **irrelated and contrast rationales** in Appendix R.
  - (iv) We evaluate **the quality of the generated rationales** in Appendix S.
- To further improve KALE's performance, we explore **combining KALE with SFT** in a sequential manner in Appendix T.

### 5.1 EXPERIMENT SETUPS

**Implementations and Benchmarks.** We apply **six** open-source LLMs scaling from 7B to 32B, including LlaMA3 8B (Team, 2024c), Mistral 7B (Jiang et al., 2023a), Qwen2.5-32B (Team, 2024d), Gemma2 9B (Team et al., 2024), OLMOE 7B (Muennighoff et al., 2024), and Orca2 7B (Mitra et al., 2023). Experiments where the model size is under 32B are conducted on 8 NVIDIA SXM A100 80G GPUs, while for models with 32B size are on 16 NVIDIA SXM A100 80G GPUs. For each benchmark, the reported performance stems from a model fine-tuned exclusively on the specific dataset's training data. We apply tasks for **logical reasoning**, including AbsR (Xiong et al., 2024), Commonsense (denote by Common) (Xiong et al., 2023), and Big Bench Hard (BBH) (Suzgun et al., 2023), **reading comprehension** including RACE-H and RACE-M (Lai et al., 2017), and **natural language understanding** including MMLU (Hendrycks et al.), ARC-c, and ARC-e (Clark et al., 2018). We use **accuracy** as the evaluation metric. **More details are in Appendix E.**

**Baseline Methods.** We compare **thirteen** baselines: (i) **Vanilla:** standalone LLMs without modifications. (ii) **CoT** (Wei et al., 2022b): prompting LLMs to generate internal thoughts. (iii) **Think-on-Graph (TOG)** (Sun et al.): applying iterative beam search to enhance LLMs' reasoning ability. (iv) **StructGPT** (Jiang et al., 2023b): proposing iterative reading-then-reasoning based on structured data. (v) **GraphRAG** (Edge et al., 2024): integrating KG traversal to retrieve structured relationships. (vi)

Table 1: Results of our KALE using LlaMA3 8B, Mistral 7B, and Qwen2.5 32B as backbone models (for more results of different backbone models, please see Appendix L). We **bold** the best results and underline the suboptimal results for each backbone model.

| Backbone | Category | Method | AbsR | ARC-c | ARC-e | Common | MMLU | BBH | RACE-h | RACE-m |
|---|---|---|---|---|---|---|---|---|---|---|
| LlaMA3 8B | Prompt-based | Vanilla | 62.68 | 66.79 | 69.90 | 58.72 | 55.88 | 46.54 | 53.35 | 57.02 |
| | | CoT | 63.15 | 71.67 | 69.34 | 54.67 | 56.83 | 48.55 | 54.31 | 57.02 |
| | Retrieval-based | TOG | 65.98 | 69.93 | 72.23 | 61.87 | 56.97 | 48.81 | 58.60 | 59.80 |
| | | StructGPT | 65.35 | 70.50 | 73.34 | 62.32 | 58.87 | 49.58 | 60.03 | 60.86 |
| | | GraphRAG | 75.83 | 74.83 | 75.76 | 61.51 | 57.28 | 55.83 | 60.89 | 69.57 |
| | SFT-based | SFT | 67.77 | 68.23 | 71.74 | 59.79 | 58.00 | 45.39 | 56.17 | 58.91 |
| | | SDFT | 76.15 | 74.91 | 71.44 | 62.24 | 58.78 | 52.37 | 56.88 | 61.03 |
| | | DMT | 74.57 | 70.82 | 72.84 | 61.43 | 59.11 | 50.14 | 61.46 | 60.64 |
| | | MeanLearn | 71.09 | 72.53 | 74.53 | 63.39 | 58.79 | 50.61 | 60.03 | 61.84 |
| | | KG-SFT | 78.20 | 73.12 | 79.55 | 63.09 | 58.79 | 53.68 | 64.98 | 62.47 |
| | Augmented-based | STaR | 69.95 | 71.50 | 70.99 | 58.20 | 53.41 | 50.07 | 61.21 | 64.32 |
| | | AugGPT | 64.45 | 72.22 | 75.29 | 55.12 | 56.82 | 51.90 | 59.21 | 60.16 |
| | | GPT3Mix | 68.27 | 70.57 | 74.24 | 61.33 | 57.79 | 53.92 | 61.03 | 62.67 |
| | | KALE (ours) | **83.62** | **81.23** | **86.45** | **65.69** | **63.27** | **57.33** | **68.61** | **74.12** |
| Mistral 7B | Prompt-based | Vanilla | 62.35 | 52.05 | 68.31 | 39.15 | 37.43 | 28.68 | 50.14 | 55.92 |
| | | CoT | 67.18 | 58.45 | 66.08 | 36.94 | 43.57 | 31.60 | 55.15 | 58.98 |
| | Retrieval-based | TOG | 64.60 | 57.25 | 70.41 | 50.78 | 41.35 | 31.29 | 52.20 | 56.96 |
| | | StructGPT | 65.17 | 57.94 | 69.28 | 46.11 | 44.94 | 32.98 | 55.69 | 60.10 |
| | | GraphRAG | 68.26 | 57.76 | 71.93 | 48.24 | 45.53 | 35.12 | 57.15 | 62.60 |
| | SFT-based | SFT | 68.48 | 55.89 | 71.55 | 44.14 | 48.86 | 34.90 | 57.09 | 61.00 |
| | | SDFT | 73.82 | 61.01 | 73.61 | 51.84 | 52.19 | 34.97 | 64.32 | 65.53 |
| | | DMT | 73.22 | 57.00 | 72.85 | 49.71 | 50.49 | 35.89 | 61.64 | 64.42 |
| | | MeanLearn | 70.97 | 64.42 | 71.55 | 47.83 | 50.95 | 35.58 | 61.06 | 64.42 |
| | | KG-SFT | 72.39 | 65.96 | 72.94 | 54.55 | 52.10 | 34.20 | 61.15 | 63.37 |
| | Augmented-based | STaR | 70.02 | 57.85 | 74.53 | 49.80 | 41.02 | 35.89 | 55.09 | 59.12 |
| | | AugGPT | 65.28 | 59.73 | 72.77 | 48.24 | 40.24 | 33.21 | 57.75 | 59.96 |
| | | GPT3Mix | 59.72 | 61.69 | 71.93 | 53.97 | 39.84 | 36.04 | 56.75 | 60.10 |
| | | KALE (ours) | **76.90** | **71.59** | **77.95** | **59.05** | **54.21** | **39.26** | **67.98** | **70.06** |
| Qwen2.5 32B | Prompt-based | Vanilla | 66.35 | 75.09 | 80.10 | 65.52 | 80.47 | 69.01 | 71.47 | 76.95 |
| | | CoT | 68.72 | 76.79 | 82.07 | 66.34 | 81.65 | 69.79 | 73.58 | 77.64 |
| | Retrieval-based | TOG | 74.64 | 80.55 | 84.13 | 68.63 | 83.27 | 72.09 | 74.12 | 78.34 |
| | | StructGPT | 74.17 | 82.43 | 83.29 | 71.58 | 83.41 | 71.93 | 75.56 | 77.92 |
| | | GraphRAG | 75.24 | 80.20 | 84.18 | 69.00 | 84.85 | 73.20 | 75.84 | 77.37 |
| | SFT-based | SFT | 72.03 | 79.61 | 83.33 | 67.89 | 82.82 | 70.40 | 73.99 | 79.39 |
| | | SDFT | 73.34 | 80.80 | 84.30 | 71.25 | 84.13 | 71.01 | 74.59 | 80.71 |
| | | DMT | 75.24 | 81.48 | 86.07 | 70.43 | 85.17 | 73.62 | 75.72 | 80.01 |
| | | MeanLearn | 71.09 | 76.37 | 84.18 | 69.12 | 83.61 | 72.85 | 74.64 | 81.82 |
| | | KG-SFT | 78.91 | 78.41 | 84.13 | 69.62 | 84.26 | 72.39 | 74.80 | 80.43 |
| | Augmented-based | STaR | 72.99 | 83.87 | 84.60 | 69.21 | 85.24 | 73.16 | 76.30 | 80.43 |
| | | AugGPT | 78.91 | 84.47 | 86.27 | 68.96 | 85.04 | 71.93 | 77.16 | 81.89 |
| | | GPT3Mix | 80.10 | 85.23 | 87.33 | 69.53 | 85.69 | 73.47 | 76.21 | 80.77 |
| | | KALE (ours) | **91.82** | **89.93** | **94.90** | **75.02** | **88.59** | **77.91** | **81.76** | **86.70** |

**SFT** (Wei et al., 2022a): standalone SFT process. (vii) **Self-Distillation Fine-Tuning (SDFT)** (Yang et al., 2024): guiding fine-tuning with a dataset generated by model itself. (viii) **Dual-stage Mixed Fine-tuning (DMT)** (Dong et al., 2023): achieving a balance between general and specialized ability. (ix) **MeanLearn** (Xiong et al., 2024): teaching LLMs to leverage generic facts. (x) **KG-SFT** (Chen et al., 2025): utilizing KGs to filter SFT data to enhance LLMs' ability. (xi) **Self-Taught Reasoner (STaR)** (Zelikman et al., 2024): generating a rationale dataset from a few initials iteratively. (xii) **AugGPT** (Dai et al., 2023): using an LLM to rephrase questions in original data. (xiii) **GPT3Mix** (Yoo et al., 2021): prompting an LLM to generate similar questions in the SFT data.

## 5.2 MAIN RESULTS

We conduct experiments using three representative LLMs with varied scales: LlaMA3 8B, Mistral 7B, and Qwen2.5 32B in Table 1. We also provide more results for other *three* different open-source LLMs in Table 13, Figures 9, and 10 in Appendix L to demonstrate the versatility of KALE across different LLMs. From Table 1, we observe that KALE consistently and significantly outperforms other state-of-the-art baselines across three LLMs by a substantial margin. **Notably, for the logical reasoning task using the AbsR benchmark, KALE achieves a maximum accuracy improvement of** 11.72% **when utilizing Qwen2.5** 32**B as the backbone model.** We also observe that traditional SFT-based and or data augmentation methods yield marginal improvements on downstream tasks,

Table 2: Results of the ablation study of KALE, using LlaMA3 8B, Mistral 7B, and Qwen2.5 32B as backbones (We provide more results for the other three backbones in Appendix N).

| Backbone | Method | AbsR | ARC-c | ARC-e | Common | MMLU | BBH | RACE-h | RACE-m |
|---|---|---|---|---|---|---|---|---|---|
| LlaMA3 8B | KALE$_{\text{w/o KI}}$ | 78.91$_{\downarrow 4.71}$ | 76.79$_{\downarrow 4.44}$ | 81.65$_{\downarrow 4.80}$ | 65.52$_{\downarrow 0.17}$ | 60.09$_{\downarrow 3.18}$ | 55.21$_{\downarrow 2.12}$ | 64.15$_{\downarrow 4.46}$ | 69.50$_{\downarrow 4.62}$ |
| | KALE$_{\text{w/o KA}}$ | 73.93$_{\downarrow 9.69}$ | 75.26$_{\downarrow 5.97}$ | 78.70$_{\downarrow 7.75}$ | 63.06$_{\downarrow 2.63}$ | 60.74$_{\downarrow 2.53}$ | 53.68$_{\downarrow 3.65}$ | 60.03$_{\downarrow 8.58}$ | 64.76$_{\downarrow 9.36}$ |
| | **KALE** | **83.62** | **81.23** | **86.45** | **65.69** | **63.27** | **57.33** | **68.61** | **74.12** |
| Mistral 7B | KALE$_{\text{w/o KI}}$ | 71.09$_{\downarrow 5.81}$ | 66.30$_{\downarrow 5.29}$ | 65.45$_{\downarrow 12.50}$ | 57.58$_{\downarrow 1.47}$ | 52.58$_{\downarrow 1.63}$ | 36.81$_{\downarrow 2.45}$ | 64.95$_{\downarrow 3.03}$ | 66.85$_{\downarrow 3.21}$ |
| | KALE$_{\text{w/o KA}}$ | 65.64$_{\downarrow 11.26}$ | 63.91$_{\downarrow 7.68}$ | 63.05$_{\downarrow 14.90}$ | 56.84$_{\downarrow 2.21}$ | 49.05$_{\downarrow 5.16}$ | 35.74$_{\downarrow 3.52}$ | 62.78$_{\downarrow 5.20}$ | 64.00$_{\downarrow 6.06}$ |
| | **KALE** | **76.90** | **71.59** | **77.95** | **59.05** | **54.21** | **39.26** | **67.98** | **70.06** |
| Qwen2.5 32B | KALE$_{\text{w/o KI}}$ | 87.32$_{\downarrow 4.50}$ | 87.03$_{\downarrow 2.90}$ | 89.98$_{\downarrow 4.92}$ | 71.01$_{\downarrow 4.01}$ | 86.87$_{\downarrow 1.72}$ | 75.15$_{\downarrow 2.76}$ | 78.04$_{\downarrow 3.72}$ | 83.57$_{\downarrow 3.13}$ |
| | KALE$_{\text{w/o KA}}$ | 82.94$_{\downarrow 8.88}$ | 85.32$_{\downarrow 4.61}$ | 88.38$_{\downarrow 6.52}$ | 70.43$_{\downarrow 4.59}$ | 84.91$_{\downarrow 3.68}$ | 76.69$_{\downarrow 1.22}$ | 77.82$_{\downarrow 3.94}$ | 82.94$_{\downarrow 3.76}$ |
| | **KALE** | **91.82** | **89.93** | **94.90** | **75.02** | **88.59** | **77.91** | **81.76** | **86.70** |

particularly when applied to larger and more powerful LLMs (e.g., only a 1.39% improvement on the BBH benchmark when using Qwen2.5 32B). In contrast, KALE delivers a consistent and significant improvement on larger LLMs. This indicates that as LLMs scale up and become more capable, SFT-based methods that focus on learning input-output patterns or data augmentation methods that merely increase the quantity of data are suboptimal to further enhance LLMs. In contrast, **KALE proposes a way to improve LLMs' ability to manipulate knowledge, which achieves better results on larger LLMs significantly.**

## 5.3 ABLATION STUDY

To investigate the contribution of each component within KALE, we conduct ablation experiments on the entire framework. We present the ablation results of KALE using LlaMA3 8B, Mistral 7B, and Qwen2.5 32B as the three representative backbone models of different structures and scales in Table 2. **More Results using the other three backbone models are in Table 13 in Appendix N.**

**Ablation on Rationale Generation** We denote KALE *without* **K**nowledge-**I**nduced (KI) data synthesis as KALE$_{\text{w/o KI}}$. That is, we do not utilize our proposed multi-path A* algorithm to provide reasoning paths for each Q&A pair. Instead, we directly input the Q&A pair and prompt the LLM to generate rationales. As shown in Table 2, we observe that using rationales directly generated by prompting LLMs without reasoning paths leads to a performance degradation. Notably, when using Mistral 7B as the backbone model, the degradation on the ARC-e dataset reaches 12.50%. This demonstrates that the extracted reasoning paths effectively capture the thought process from question to answer, which contributes to the generation of higher-quality rationales.

**Ablation on KL Divergence** We denote KALE without the **K**nowledge-**A**ware (KA) fine-tuning as KALE$_{\text{w/o KA}}$. That is, we directly apply the rationale data generated through the KG to LLM using Cross-Entropy loss as the objective function. We observe that aligning LLMs' outputs with and without rationales using cross-entropy does not achieve satisfactory results. Specifically, when using Mistral 7B as the backbone on the ARC-e dataset, KALE$_{\text{w/o KA}}$ results in a 14.90% degradation. This demonstrates the effectiveness of the KL divergence for better knowledge manipulation performance.

## 5.4 CASE STUDY

**Known&incorrect Phenomenon** As illustrated in Figure 1, LLMs via SFT still exhibit the known&incorrect phenomenon. We provide a detailed analysis of six different LLMs after SFT and KALE (please refer to Appendix O for results of other baselines). We use the known fact checking process in Figure 1 (please refer to Appendix J for prompt details) to categorize LLMs' responses **under the precondition that LLMs already possesses relevant knowledge**: (i) *Known&correct*: LLMs possess the knowledge and correctly answers the question, which indicates a successful knowledge manipulation. (ii) *Known&incorrect*: LLMs possess the knowledge yet cannot correctly answer questions, which indicates an inflexible knowledge manipulation. As shown in the left part of Figure 3, we observe that SFT models often exhibit the *known&incorrect* phenomenon. More than 25% of the questions are cases where LLM possesses the knowledge to answer but cannot provide correct responses. In OLMOE 7B, it reaches 44.1%. In contrast, LLMs via KALE demonstrate excellent knowledge manipulation ability, with less than 10% *known&incorrect* issues of questions

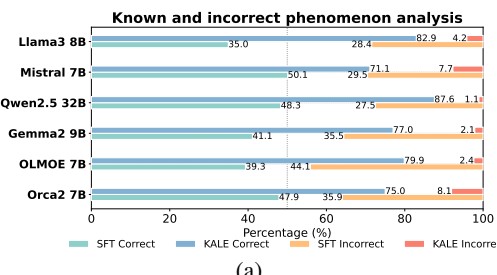 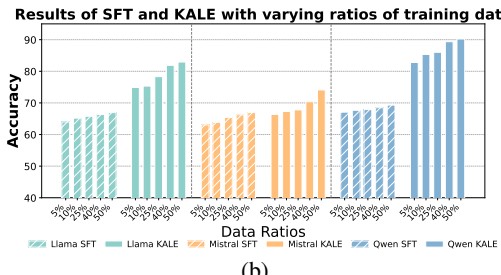

(a)              (b)

Figure 3: We illustrate two case study results to provide more insights into our KALE. **(i)** *Known&incorrect* **phenomenon analysis:** following the known fact checking in Figure 1, we collect cases where LLMs possess the knowledge to answer and analyze the ratios of correct and incorrect answers provided by LLMs, denoted as *known&correct* and *known&incorrect*. **(ii) Ratios of augmented rationales**: by setting the data augmentation ratio from 5% to 50%, we explore the differences between KALE and the SFT under varying data scales. We provide results of LlaMA3 8B, Mistral 7B, and Qwen2.5 32B as the backbones as examples, with more results in Appendix P.

across all LLMs. Notably, for the Qwen2.5 32B model, this proportion dropped to as low as 1.1%. These results indicate that KALE effectively enhances LLM's knowledge manipulation ability.

**Ratios of Augmented Rationales** In real-world applications, data acquisition in certain domains can be particularly challenging due to privacy concerns, security restrictions, etc. (Rodríguez-Mazahua et al., 2016). Therefore, we investigate KALE and SFT under limited training data scenarios. Taking the AbsR dataset as an example, by setting the training data ratio from only 5% to 50%, we provide the results in the right part of Figure 3. We find that KALE consistently outperforms SFT methods across all levels of augmented rationales. Moreover, this improvement becomes more significant on Qwen-2.5 32B, which also demonstrates that our KALE is highly effective on more powerful LLMs. **This highlights the significant potential of KALE for low-data, real-world applications.**

## 6 CONCLUSION

**Conclusion** In this paper, we propose a novel **K**nowledge-**A**ware **LE**arning (KALE) framework to enable better knowledge manipulation ability of LLMs. Specifically, KALE consists of (i) a knowledge-induced data synthesis method to generate high-quality rationales for each Q&A pair through a structured knowledge graph, and (ii) a knowledge-aware fine-tuning paradigm to enhance the knowledge manipulation ability of LLMs. Extensive experiments on **eight** benchmarks and **six** open-source models across different scales, ranging from 7B to 32B, demonstrate the superiority of our KALE, delivering significant, consistent, and generalizable improvements.[3]

**Limitations and Future Work** We consider a few limitations and future directions. (i) Current KALE relies on a structured Q&A dataset to facilitate knowledge-induced data synthesis. For cases where a Q&A dataset is not available, users can consider employing GPT-4o or other LLMs to transform a raw corpus into a structured Q&A. We think applying KALE directly to raw data is a promising direction. (ii) When generating reasoning paths, multi-path A* algorithm is a hard-match approach. Obtaining vectorized embeddings for similarity-based matching is also an optimization direction. (iii) In multi-path A*, we empirically sample k anchor nodes for distance estimation. Finer entity-specific selection (e.g., a neural decision module) may yield better results. (iv) Current KALE relies on an available KG, which may constrain its applicability in domains where specialized KGs are scarce. Meanwhile, many areas—including the medical domain—already benefit from community-maintained Wikidata, whose ongoing expansion enhances its value for diverse applications. We are further encouraged by advances in the KG community that target automatic construction of domain-specific KGs: methods like SAC-KG (Chen et al., 2024) show promise in building high-quality KGs. Such approaches are pivotal for extending KALE to domains where mature KGs are not available.

---

[3]More discussions on KALE can be found in Appendix U.

## ETHICS STATEMENT

This work adheres to the ICLR Code of Ethics. Our study does not involve human subjects or personally identifiable information (PII), and we did not collect new sensitive data. All datasets are publicly available under their respective licenses, and third-party resources are credited. We report methods and results transparently, consider potential risks such as misuse or bias amplification, and do not recommend deployment in high-stakes settings without additional safety assessment.

## REPRODUCIBILITY STATEMENT

We provide details to facilitate replication: dataset names and versions, preprocessing steps, model/configuration, training schedules, and evaluation protocols. All hyperparameters are listed in the appendix; scripts and configs are included in the anonymous supplementary materials. For theoretical or algorithmic components, assumptions and full proofs are provided in the appendix. These pointers collectively enable independent reproduction of our results.

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

## A MORE RELATED WORKS

### A.1 LARGE LANGUAGE MODELS

The advent of pre-trained language models has fundamentally transformed the landscape of natural language processing (NLP), marking a significant paradigm shift in how language understanding and generation tasks are approached. The pioneering work of the GPT series (Radford et al., 2018; Brown et al., 2020b;b) introduced the concept of unsupervised pre-training followed by task-specific fine-tuning, demonstrating the effectiveness of leveraging large-scale unlabeled text corpora. This approach was further refined by BERT (Devlin et al., 2018), which introduced bidirectional context encoding through the masked language modeling objective, achieving state-of-the-art results across a wide range of NLP benchmarks. Subsequent advancements, such as RoBERTa (Liu et al., 2019), optimized the pre-training process by removing the next sentence prediction objective and training with larger batches and more data, leading to improved performance. Megatron-LM then (Shoeybi et al., 2019) showcased the scalability of these models, leveraging model parallelism to train significantly larger architectures. More recently, the field has witnessed the emergence of LLMs that have pushed the boundaries of what is possible in NLP. Models such as LlaMA3 (Team, 2024c; 2023b), ChatGPT (OpenAI, 2020), GPT-4.5 (OpenAI, 2024), PaLM (Team, 2022), Gemini (Team, 2023a), Claude3 (Team, 2024a), and Deepseek V3 (Team, 2024b) have demonstrated remarkable abilities in both few-shot and zero-shot learning scenarios (Brown et al., 2020a). These models, often comprising hundreds of billions of parameters, have been pre-trained on diverse and extensive benchmarks, enabling them to generalize across a wide array of tasks with minimal or no task-specific fine-tuning. The evolution from earlier models like GPT and BERT to the current generation of LLMs underscores the importance of scale and the effectiveness of pre-training on large corpora. These advancements have not only improved performance on traditional NLP tasks but have also enabled new applications and capabilities, such as conversational agents (Chen et al., 2024), code generation (Zan et al., 2023; Liu et al., 2025), and complex reasoning tasks (Lv et al., 2024). The continued development and refinement of these models promise to further enhance their utility and impact across various domains.

### A.2 CLASSIC TEXT DATA AUGMENTATION METHODS

Data augmentation has long been a foundational research area in natural language processing (NLP), aimed at enhancing the quality and diversity of training data to improve model generalization and performance. Traditional data augmentation techniques have predominantly focused on character-level and word-level modifications. An example is Easy Data Augmentation (EDA) (Wei & Zou, 2019), which employs straightforward yet effective strategies such as random insertion, random swapping, random deletion, and synonym replacement to introduce variability into the benchmark (Belinkov & Bisk, 2018; Coulombe, 2018; Wang et al., 2023b). These methods, while computationally efficient, are often limited in their ability to generate semantically coherent and contextually rich variations, particularly at higher linguistic levels such as sentences or documents.

### A.3 CHAIN-OF-X APPROACHES IN LLMS

The ability of LLMs to decompose complex problems into a series of intermediate steps and generate internal reasoning processes, known as Chain-of-Thought (CoT) prompting (Wei et al., 2022b), represents a significant advancement in enhancing their reasoning capabilities. The CoT approach emulates human problem-solving strategies by breaking down intricate problems into smaller, more manageable components. This step-by-step reasoning process allows LLMs to focus on each segment individually, reducing errors and improving logical coherence in their responses (Wang et al.). Moreover, CoT explicitly encourages models to articulate their thought processes, which not only facilitates debugging and refinement of the model's reasoning but also significantly enhances the interpretability of its outputs. As a result, responses generated through CoT are often more accurate, logically consistent, and contextually relevant compared to those produced by models that directly generate final answers without revealing intermediate cognitive steps. The success of CoT has inspired a series of follow-up works that extend its principles to other chain-of-X methods, further broadening its applicability and effectiveness. For instance, chain-of-explanation (Huang et al., 2023) focuses on generating detailed explanations to justify the reasoning process, while chain-of-knowledge (Wang et al., 2023a) emphasizes the integration of external knowledge to enrich the model's responses.

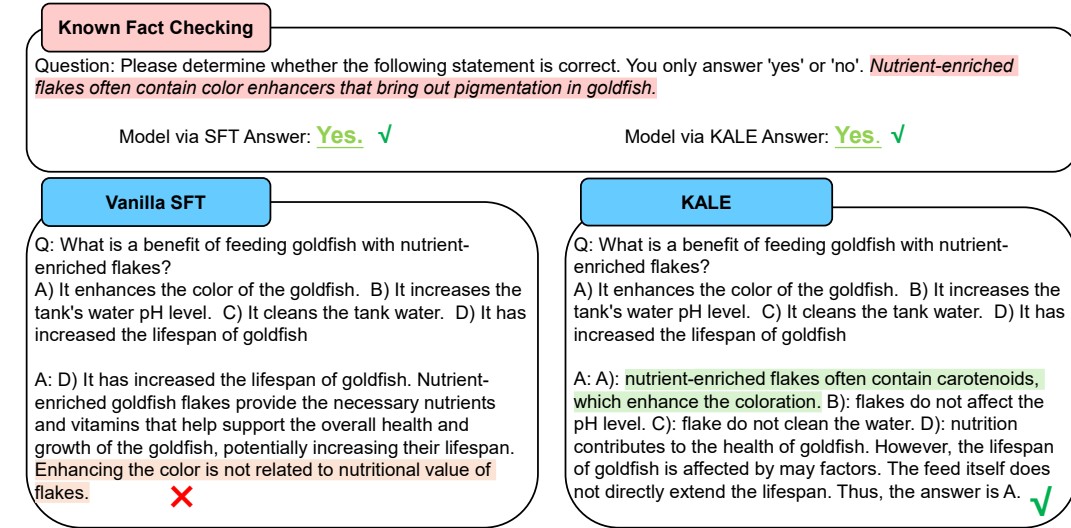

Figure 4: An example of *known&incorrect* phenomenon comparison using LlaMA3 8B as the backbone model.

More recently, chain-of-verification (Dhuliawala et al., 2023) has been proposed to enhance the reliability of LLMs by prompting them to draft initial responses, plan verification questions, answer those questions, and generate a final verified response. This iterative verification process reduces the likelihood of misunderstandings or errors in the model's reasoning. Another notable extension is Chain-of-Knowledge (Zhang et al., 2024), which elicits LLMs to generate explicit pieces of knowledge evidence in the form of structured triples. This approach is inspired by human cognitive behaviors, where individuals often draw mind maps or knowledge maps as reasoning evidence before addressing complex questions. By structuring knowledge in this way, LLMs can better organize and utilize information, leading to more informed and accurate responses.

## B    MORE CASES OF THE *Known&incorrect* PHENOMENON

In Figure 1, we present a comparative analysis of the *known&incorrect* phenomenon of models fine-tuned after SFT and KALE, using Mistral-7B as the backbone model. In this section, we further extend the investigation by providing more *known&incorrect* phenomenon comparisons **across LlaMA3 8B, Qwen2.5 32B, Gemma2 9B, OLMOE 7B, and Orca2 7B** on various domains to comprehensively demonstrate the efficacy of our proposed KALE. **As illustrated in Figures 4, 5, 6, 7, and 8**, we still find that that models fine-tuned after SFT still exhibit the *known&incorrect* phenomenon, wherein the models cannot properly recall and apply acquired knowledge to answer correctly despite possessing the relevant knowledge. In contrast, LLMs fine-tuned after KALE demonstrate a better ability to effectively manipulate relevant knowledge to generate correct answers. These results also demonstrate that our KALE effectively strengthens LLMs' knowledge manipulation ability.

## C    PROOF OF ADMISSIBILITY OF THE PROPOSED MULTI-PATH A* ALGORITHM

In this section, we show that our proposed heuristic estimated cost in equation 2 is admissible,i.e., $h(\mathbf{e}) \leq \mathrm{dist}(\mathbf{e}, \mathbf{e}_g)$ for any node $\mathbf{e}$, which means that our proposed multi-path A* algorithm can find the best solution. We resort to the triangle inequality property of the distance metric $\mathrm{dist}(x, y)$. For any three nodes $A, B, C$, the triangle inequality states:

$$\mathrm{dist}(A, C) \leq \mathrm{dist}(A, B) + \mathrm{dist}(B, C) \tag{5}$$

**Known Fact Checking**

Question: Please determine whether the following statement is correct. You only answer 'yes' or 'no'. *Nutrient-enriched flakes often contain color enhancers that bring out pigmentation in goldfish.*

Model via SFT Answer: **Yes.** √          Model via KALE Answer: **Yes.** √

**Vanilla SFT**

Q: What is a benefit of feeding goldfish with nutrient-enriched flakes?
A) It enhances the color of the goldfish. B) It increases the tank's water pH level. C) It cleans the tank water. D) It has increased the lifespan of goldfish

A: D) It has increased the lifespan of goldfish. Nutrient-enriched goldfish flakes provide the necessary nutrients and vitamins that help support the overall health and growth of the goldfish, potentially increasing their lifespan. Enhancing the color is not related to nutritional value of flakes. ✗

**KALE**

Q: What is a benefit of feeding goldfish with nutrient-enriched flakes?
A) It enhances the color of the goldfish. B) It increases the tank's water pH level. C) It cleans the tank water. D) It has increased the lifespan of goldfish

A: A): nutrient-enriched flakes often contain carotenoids, which enhance the coloration. B): flakes do not affect the pH level. C): flake do not clean the water. D): nutrition contributes to the health of goldfish. However, the lifespan of goldfish is affected by may factors. The feed itself does not directly extend the lifespan. Thus, the answer is A. √

Figure 5: An example of *known&incorrect* phenomenon comparison using Qwen2.5 32B as the backbone model.

**Known Fact Checking**

Question: Please determine whether the following statement is correct. You only answer 'yes' or 'no'. *Bananas grow on plants in a way that their fruit points upward, defying gravity. This is known as "negative geotropism," where the fruits grow away from the pull of gravity.*

Model via SFT Answer: **Yes.** √          Model via KALE Answer: **Yes.** √

**Vanilla SFT**

Q: Which direction do bananas grow on the plant?
A) Downward. B) Upward. C) Horizontally. D) Underground.

A: Bananas grow in bunches from a central stem. The bunch itself emerges from the plant's "pseudostem," which is actually a collection of tightly wrapped leaf sheaths, not a true woody stem like a tree. Considering the weight of the fruit and the structure of the plant, we can deduce that the growth direction is downward. ✗
Answer: A) Downward.

**KALE**

Q: Which direction do bananas grow on the plant?
A) Downward. B) Upward. C) Horizontally. D) Underground.

A: Bananas grow on a plant called a banana tree, which is a large herb. The fruit grows in clusters called bunches, and the bananas themselves grow upward. This happens because bananas have a natural tendency for "negative geotropism", meaning they grow against the force of gravity, seeking sunlight. √
Answer: B) Upward.

Figure 6: An example of *known&incorrect* phenomenon comparison using Gemma2 9B as the backbone model.

Let us consider an arbitrary landmark $\alpha_i$ from the set $\{\alpha_i\}_{i=1}^k$. Applying the triangle inequality with $A = \alpha_i$, $B = \mathbf{e}$, and $C = \mathbf{e}_g$, we have:

$$\text{dist}(\alpha_i, \mathbf{e}_g) \leq \text{dist}(\alpha_i, \mathbf{e}) + \text{dist}(\mathbf{e}, \mathbf{e}_g) \tag{6}$$

Rearranging Equation equation 6, we obtain:

$$\text{dist}(\alpha_i, \mathbf{e}_g) - \text{dist}(\alpha_i, \mathbf{e}) \leq \text{dist}(\mathbf{e}, \mathbf{e}_g) \tag{7}$$

Now, let $X_i = \text{dist}(\alpha_i, \mathbf{e}_g) - \text{dist}(\alpha_i, \mathbf{e})$. The term in the heuristic function involving $\alpha_i$ is $[X_i]^+ = \max(X_i, 0)$. We consider two cases for the value of $X_i$:

1. **Case 1:** $X_i < 0$, due to the non-negative property of the distance, we have:

$$[\text{dist}(\alpha_i, \mathbf{e}_g) - \text{dist}(\alpha_i, \mathbf{e})]^+ = 0 \leq \text{dist}(\mathbf{e}, \mathbf{e}_g)$$

**Known Fact Checking**

Question: Please determine whether the following statement is correct. You only answer 'yes' or 'no'. *When onions are cut, their cells rupture and release enzymes that convert sulfur-containing compounds into volatile sulfuric gases. These gases react with the moisture in the eyes, forming mild sulfuric acid, which irritates the eyes and triggers tear production.*

Model via SFT Answer: **Yes.** √          Model via KALE Answer: **Yes.** √

**Vanilla SFT**

Q: Why do onions make you cry?
A) Acidic juice irritates eyes. B) Enzymes release sulfuric gas. C) Pollen triggers allergies. D) Natural defense against cutting.

A: Onions make you cry primarily because of their acids. The acids are released when you cut the onion, and these acids quickly diffuses into the air and into your eyes, which are moist, causing a burning sensation and tearing. This is a natural defense mechanism of the onion to deter predators from eating it.
Answer: A) Acidic juice irritates eyes.          ✗

**KALE**

Q: Why do onions make you cry?
A) Acidic juice irritates eyes. B) Enzymes release sulfuric gas. C) Pollen triggers allergies. D) Natural defense against cutting.

A : When an onion is cut, its cells are damaged. This damage allows enzymes within the onion to mix with sulfur-containing amino acids. Alliinase helps synthase acts on these sulfenic acids to produce a sulfur-containing gas. When it reaches eyes, it stimulates the lachrymal glands, causing them to produce tears to flush away the irritant.
Answer: B) Enzymes release sulfuric gas.          √

Figure 7: An example of *known&incorrect* phenomenon comparison using OLMOE 7B as the backbone model.

**Known Fact Checking**

Question: Please determine whether the following statement is correct. You only answer 'yes' or 'no'. *Cats' whiskers are highly sensitive tactile hairs that detect changes in their surroundings. Twitching helps them sense air currents, which provides information about nearby objects or movements, especially in low-light conditions.*

Model via SFT Answer: **Yes.** √          Model via KALE Answer: **Yes.** √

**Vanilla SFT**

Q: Why do cats' whiskers twitch?
A) To measure the width of spaces. B) To detect air currents. C) For balance. D) To communicate emotions.

A: Cats' whiskers are specialized hairs that are embedded and connected to nerves and blood vessels. They serve multiple sensory functions. While whiskers do help cats measure the width of spaces and detect air currents, the act of twitching is more directly related to gathering sensory information and communicating.
Answer: D) To communicate emotions.          ✗

**KALE**

Q: Why do cats' whiskers twitch?
A) To measure the width of spaces. B) To detect air currents. C) For balance. D) To communicate emotions.

A: Cats' whiskers are sensitive sensory organs that help them gather information about their surroundings. Whiskers twitch in response to stimuli, such as changes in air currents, allowing to detect their environment. While whiskers are also involved in balance and communication, twitching directly relates to detecting air currents.
Answer: B) To detect air currents.          √

Figure 8: An example of *known&incorrect* phenomenon comparison using Orca2 7B as the backbone model.

2. **Case 2:** $X_i \geq 0$, from Eq.equation 7, we know that $\text{dist}(\alpha_i, \mathbf{e}_g) - \text{dist}(\alpha_i, \mathbf{e}) \leq \text{dist}(\mathbf{e}, \mathbf{e}_g)$. Therefore:

$$[\text{dist}(\alpha_i, \mathbf{e}_g) - \text{dist}(\alpha_i, \mathbf{e})]^+ \leq \text{dist}(\mathbf{e}, \mathbf{e}_g)$$

In both cases, for any anchor $\alpha_i$ ($1 \leq i \leq k$), we have shown that:

$$[\text{dist}(\alpha_i, \mathbf{e}_g) - \text{dist}(\alpha_i, \mathbf{e})]^+ \leq \text{dist}(\mathbf{e}, \mathbf{e}_g) \quad (8)$$

The heuristic function $h(\mathbf{e})$ is defined as the maximum of these terms over all $i$:

$$h(\mathbf{e}) = \max_{1 \leq i \leq k} \left[\text{dist}(\alpha_i, \mathbf{e}_g) - \text{dist}(\alpha_i, \mathbf{e})\right]^+$$

Since each term $[\text{dist}(\alpha_i, \mathbf{e}_g) - \text{dist}(\alpha_i, \mathbf{e})]^+$ is less than or equal to $\text{dist}(\mathbf{e}, \mathbf{e}_g)$, their maximum must also be less than or equal to $\text{dist}(\mathbf{e}, \mathbf{e}_g)$. Thus,

$$h(\mathbf{e}) \leq \text{dist}(\mathbf{e}, \mathbf{e}_g) \quad (9)$$

This inequality holds for any node e. **Therefore, our proposed multi-path A\* algorithm is admissible, which means that for any node, our proposed multi-path A\* algorithm can find the best solution.**

## D  PSEUDO CODE OF THE PROPOSED MULTI-PATH A\*

In Section 4.1, we introduce our multi-path A\* algorithm, which efficiently extracts inference paths from question entities to answer entities. Here, we provide the algorithm pseudo code in Algorithm 1.

---

**Algorithm 1** Pseudo code for Multi-path A\* algorithm

---

**Input:** Start node $\mathbf{e}_q$, target node $\mathbf{e}_a$, maximum number of paths $m$ and maximum search depth $d$

1: Initialize priority queue $\mathcal{Q}$ with $(f(\mathbf{e}), g(\mathbf{e}), \mathbf{e}, p_{\mathbf{e}}^i)$
2: Initialize reasoning path and visited list $\mathcal{P}, \mathcal{V} \leftarrow [], []$
3: **while** $\mathcal{Q} \neq \emptyset$ and $|\mathcal{P}| < m$ **do**
4:      Dequeue the element with the smallest $f(\mathbf{e})$ from $\mathcal{Q}$
5:      Append $\mathbf{e}$ into $\mathcal{V}$
6:      **if** $\mathbf{e} = \mathbf{e}_a$ **then**
7:          Append $p_{\mathbf{e}}^i$ into $\mathcal{P}$
8:          **continue**     ▷ Find Reasoning Path
9:      **end if**
10:     **if** $g(\mathbf{e}) > d$ **then**
11:        **continue**     ▷ Path exceeds maximum search depth
12:     **end if**
13:     **for** each neighbor $\mathbf{n}$ of $\mathbf{e}$ **do**
14:        **if** $\mathbf{n} \in$ path **then**
15:           **continue**     ▷ Avoid cycles
16:        **end if**
17:        Obtain $g(\mathbf{n}) \leftarrow g(\mathbf{e}) + 1$
18:        Compute $f(\mathbf{n})$ and $h(\mathbf{n})$ via Equations equation 1 and equation 2
19:        Enqueue $(f(\mathbf{n}), g(\mathbf{n}), \mathbf{n}, p_{\mathbf{e}}^i + [\mathbf{n}])$ into $\mathcal{Q}$
20:     **end for**
21: **end while**
**Output:** Reasoning path list $\mathcal{P}$

---

## E  MORE DETAILS OF BENCHMARKS AND EXPERIMENT SETUPS

### E.1  IMPLEMENTATION DETAILS

In our implementation details, we conduct fine-tuning on all evaluated benchmarks across 3 epochs with a consistent batch size of 16, utilizing NVIDIA A100 GPUs (80 GB) for computational processing. The computational resources are allocated based on model scale, with 8 GPUs employed for the 7B and 8B parameter models, while the larger 32B parameter models use 16 GPUs to accommodate their increased computational demands during the fine-tuning process. For all answer entities $\mathbf{e}_a$, we choose 10 anchor entities randomly sampled from their 3-hop neighbors. To guarantee stable and reproducible results, we utilize greedy decoding by setting the temperature parameter to 0 in all experiments. The optimization process employs a peak learning rate of 3e-5, implemented in conjunction with a learning rate warmup strategy that gradually increases the learning rate over the initial 1% of training iterations to ensure stable convergence. We set the maximum truncated length as 2048 for all the benchmarks. We apply deepspeed[4] to accelerate the training process. We implement our approach based on PyTorch 2.5.1[5] and Huggingface's Transformers[6]. For the training code of KALE, we modified the training scripts based on LLaMAFactory (Zheng et al., 2024). We are committed to providing the source code of our approach, if accepted. During testing, for all models,

---

[4]https://www.deepspeed.ai/
[5]https://pytorch.org/
[6]https://github.com/huggingface/transformers

we follow MeanLearn (Xiong et al., 2024) to use the same system prompt for a fair comparison: "You are a cautious assistant. You carefully follow instructions. You are helpful and harmless, and you follow ethical guidelines and promote positive behavior. You are given a question together with a few options. You should give an explanation first and then answer the question." More details for the best performance of each task and benchmark can be seen within our code.

### E.2 BENCHMARK DETAILS

For **more details of benchmarks**, we list below all the benchmarks used in **logical reasoning, reading comprehension, and natural language understanding,** respectively, by KALE as follows. **Logical Reasoning Task** we employ AbsR (Xiong et al., 2024), Commonsense (Xiong et al., 2023), and Big Bench Hard (BBH) (Suzgun et al., 2023) as our evaluation benchmarks. Specifically, **the AbsR benchmark** was constructed using GPT-4 (gpt-4-1106-preview)[7] as the primary data annotator, following (Chen et al., 2024; Zheng et al., 2023). For each generic fact $r_i$, GPT-4 was prompted to generate samples $S_i = \{s_1^i, \ldots, s_{m_i}^i \mid 1 \leq m_i \leq 3\}$ in diverse scenarios. Each sample $s_j^i$ consists of a question $X_j^i$ with multiple options, a response $Y_j^i$ containing an answer and an explanation guided by $r_i$, and forms a triple $s_j^i = \langle X_j^i, r_i, Y_j^i \rangle$. From each sample in the training set $s_j^i$, two types of examples were derived: (i) K-example, which predicts $Y_j^i$ given $\langle X_j^i, r_i \rangle$, and (ii) R-example, which predicts $Y_j^i$ given only $X_j^i$. These examples are designed to implicitly enhance abstract reasoning in LLMs through the knowledge and reasoning pathways. In the testing set, only the R-example is provided for each sample. The statistics of the AbsR benchmark are summarized in Table 3.

Table 3: The statistics of AbsR the benchmark.

|       | Examples | Questions | Generic Facts |
|-------|----------|-----------|---------------|
| Train | 18,020   | 9,010     | 4,613         |
| Test  | 844      | 844       | 104           |

**The Commonsense benchmark** (Xiong et al., 2023) is a multiple-choice question-answering benchmark designed to evaluate the ability of LLMs to perform complex reasoning based on commonsense knowledge. Each question in the benchmark is associated with five candidate answers, only one of which is correct. The dataset spans a diverse range of domains, including everyday scenarios, social interactions, and physical phenomena, making it a comprehensive testbed for evaluating the commonsense reasoning capabilities of LLMs. We summarize the key statistics and characteristics of Commonsense in Table 4. For the BBH benchmark (Suzgun et al., 2023), it consists of a curated suite of 23 challenging tasks derived from the broader BIG-Bench benchmark (bench authors, 2023). These tasks were specifically selected because prior language model evaluations failed to surpass the average human-rater performance, making them particularly suitable for assessing the limits of current models. The tasks span a wide range of domains, including logical reasoning, mathematical problem-solving, and linguistic understanding, requiring models to demonstrate robust reasoning and contextual comprehension. BBH focuses on the importance of structured reasoning pathways in tackling complex tasks. We summarize the filtering process of BBH in Table 5.

Table 4: The statistics of the Commonsense reasoning benchmark.

| Dataset | Task Type | Size |
|---------|-----------|------|
| $\alpha$**NLI** (Bhagavatula et al., 2019) | 2 Choices | 1,507 |
| **CSQA** (Talmor et al., 2019) | 5 Choices | 1,221 |
| **COPA** (Gordon et al., 2012) | 2 Choices | 500 |
| **e-CARE** (Du et al., 2022) | 2 Choices | 2,122 |
| **Social IQa** (Sap et al., 2019) | 3 Choices | 1,935 |
| **PIQA** (Bisk et al., 2020) | 2 Choices | 1,838 |
| **StrategyQA** (Geva et al., 2021) | Yes or No | 2,290 |

**Reading Comprehension Task** We employ RACE-M (middle school level reading comprehension task) and RACE-H (high school level reading comprehension task) (Lai et al., 2017) as our benchmarks. RACE is collected from the English exams for middle and high school Chinese students in the age range between 12 to 18. RACE consists of nearly 28,000 passages and nearly 100,000 questions generated by human experts (English instructors), and covers a variety of topics that are carefully designed to evaluate the student's ability to understand and reason. The reasoning types of

---

[7]https://platform.openai.com/

Table 5: Filtering criteria to create the BIG-Bench Hard (BBH) benchmark.

| # Tasks | Criteria |
|---------|----------|
| 209 | All BIG-Bench tasks |
| 187 | - After filtering out tasks with more than three subtasks |
| 130 | - After filtering out tasks with fewer than 103 examples (3 for few-shot, 100 for evaluation) |
| 85 | - After filtering out tasks without human-rater baselines |
| 78 | - After filtering out tasks that do not use multiple-choice or exact match as the evaluation metric |
| 78 | Clean multiple-choice or exact match tasks |
| 36 | - After filtering out tasks in which the best reported model beats average reported human-rater score |
| 23 | - After filtering out extremely difficult tasks that are outside the scope of this work |
| **23** | **Remaining tasks = BIG-Bench Hard (BBH)** |

RACE include word matching, paraphrasing, single-sentence reasoning, multi-sentence reasoning, and insufficient/ambiguous. We summarize the details in Table 6.

Table 6: Statistics of the reading comprehension benchmarks, RACE-H and RACE-M. The values below the Training/Valid/Testing Set are the number of passages and questions in each dataset, respectively. Passage/Question/Option Len denotes the average length of the passages, questions, and options, respectively. Vocab size denotes the number of words in the vocabulary.

| Dataset | Training Set | Valid Set | Testing Set | Passage Len | Question Len | Option Len | Vocab Size |
|---------|-------------|-----------|-------------|-------------|--------------|------------|------------|
| RACE-M | 6,409/25,421 | 368/1,436 | 362/1,436 | 231.1 | 9.0 | 3.9 | 32,811 |
| RACE-H | 18,728/62,445 | 1,021/3,451 | 1,045/3,498 | 353.1 | 10.4 | 5.8 | 125,120 |

**Natural Language Understanding Task** For the natural language understanding task, we employ the **Massive Multitask Language Understanding (MMLU) benchmark** (Hendrycks et al.) and the ARC benchmark for evaluation. **MMLU** is a comprehensive dataset designed to assess the breadth and depth of LLMs' knowledge and problem-solving abilities. MMLU consists of 57 tasks spanning diverse domains, including STEM (Science, Technology, Engineering, and Mathematics), humanities (e.g., law, philosophy, history), social sciences (e.g., economics, sociology, psychology), and other specialized fields (e.g., medicine, finance). The dataset comprises $15,908$ questions, divided into three splits: a dev set with 5 questions per subject for few-shot evaluation, a validation set with $1,540$ questions for hyperparameter tuning, and a test set with $14,079$ questions, ensuring at least 100 test examples per subject.

The questions in MMLU are designed to require extensive world knowledge and expert-level reasoning, making it a rigorous benchmark for evaluating language models' generalization across multiple disciplines. We summarize the key statistics and characteristics of the MMLU dataset in Table 7. The **AI2 Reasoning Challenge (ARC)** benchmark (Clark et al., 2018) is a comprehensive dataset designed to assess the ability of language models to answer complex, multi-faceted science questions on scientific

Table 7: Statistics for MMLU, ARC-C, and ARC-e datasets.

| Statistics | Train | Dev | Test |
|------------|-------|-----|------|
| **MMLU** | 99,842 | 1,540 | 14,079 |
| **ARC-C** | 1,119 | 299 | 1,172 |
| **ARC-e** | 2,251 | 570 | 2,376 |

reasoning and knowledge integration capabilities. The ARC dataset consists of $7,787$ multiple-choice questions derived from grade-school-level science exams, spanning grades 3 through 9. These questions are divided into two subsets: the Easy Set (ARC-E) and the Challenge Set (ARC-C), with the latter containing $2,590$ questions that are particularly difficult and require advanced reasoning skills. The Easy Set (ARC-E) comprises $5,197$ questions that are relatively straightforward and can often be answered using basic retrieval or word co-occurrence methods. In contrast, the Challenge Set (ARC-C) includes questions that were specifically selected because they could not be correctly answered by retrieval-based algorithms (e.g., Information Retrieval Solver) or word co-occurrence methods (e.g., Pointwise Mutual Information Solver). These questions demand deeper comprehension, reasoning, and the integration of distributed knowledge across multiple sentences or concepts. Each question in the ARC dataset is presented with four answer choices, with less than 1% of questions having either three or five options. The dataset is further partitioned into training, validation, and test splits to

facilitate model development and evaluation. For instance, the Challenge Set includes $1,119$ training examples, 299 validation examples, and $1,172$ test examples. We summarize the key statistics and characteristics of the ARC dataset in Table 12.

**Medical Domain Benchmarks**  We use multiple-choice medical questions benchmarks in six languages as the representative knowledge-intensive domain, including MedQA (English and Chinese) (Jin et al., 2021), IgakuQA (Japanese) (Kasai et al., 2023), RuMedDaNet (Qiu et al., 2024), FrenchMedMCQA (Labrak et al., 2022), and Head-QA (Vilares & Gómez-Rodríguez, 2019) to provide a comprehensive understanding of our KALE. We provide the statistics of each dataset in Table 8.

Table 8: Statistical results for medical multiple-choice questions benchmarks in six languages.

| Dataset | Language | Source | Train | Test |
|---|---|---|---|---|
| MedQA | English | United States Medical Licensing Examination | 10178 | 1273 |
| MedQA | Chinese | United States Medical Licensing Examination | 27400 | 3426 |
| IgakuQA | Japanese | Japan's medical licensure exams (2018-2022) | 1590 | 199 |
| RuMedDaNet | Russian | Russian medical judgment question dataset | 1052 | 256 |
| FrenchMedMCQA | French | Professional exams for the French Pharmacy degree | 2171 | 622 |
| Head-QA | Spanish | Exams for positions in the Spanish healthcare | 2657 | 2742 |

# F  MORE RESULTS OF KALE ON KNOWLEDGE-INTENSIVE DOMAINS

In Tables 1 and 13, we present the performance of KALE across various downstream tasks. To further demonstrate the capabilities of KALE, this section provides its evaluation on several knowledge-intensive tasks. Following the same experimental setting of KG-SFT (Chen et al., 2025), we use MedQA as the benchmark using LlaMA2 7B as the backbone model. As shown in Table 9, we still observe that our proposed KALE significantly outperforms existing state-of-the-art baselines by a large margin, which also demonstrates that our KALE can effectively work under the knowledge-intensive scenarios.

Table 9: Experiment results for existing methods on knowledge-intensive domains. The results of the mentioned methods are taken from KG-SFT (Chen et al., 2025). We **bold** the best results for each dataset.

| Method | MedQA (English) | MedQA (Chinese) | IgakuQA (Russian) | RuMedDaNet (Spanish) | MedMCQA (French) | HeadQA (Japanese) | Average |
|---|---|---|---|---|---|---|---|
| Vanilla | 28.20 | 28.37 | 51.17 | 32.97 | 12.76 | 11.10 | 27.43 |
| COT | 37.65 | 39.01 | 65.23 | 40.33 | 25.08 | 23.63 | 38.48 |
| TOG | 34.27 | 28.13 | 48.42 | 35.59 | 12.47 | 19.61 | 29.75 |
| KGR | 33.15 | 26.88 | 47.52 | 34.74 | 13.39 | 17.29 | 28.83 |
| KAPING | 36.39 | 27.24 | 54.66 | 34.98 | 11.54 | 15.91 | 30.45 |
| SFT | 33.62 | 29.33 | 66.40 | 35.19 | 12.67 | 21.11 | 32.30 |
| AugGPT | 40.29 | 36.54 | 62.14 | 40.70 | 22.99 | 27.13 | 38.30 |
| GPT3Mix | 39.35 | 37.97 | 66.01 | 41.50 | 25.08 | 26.13 | 39.34 |
| KG-SFT | 41.71 | 39.31 | 68.75 | 44.40 | 28.45 | 28.14 | 41.79 |
| **KALE (ours)** | **45.89** | **42.77** | **69.81** | **45.58** | **30.39** | **28.79** | **43.53** |

Table 10: Average testing time for each sample on the AbsR dataset for each method (Unit: second)

| Backbone Models | Vanilla | CoT | TOG | StructGPT | GraphRAG | KALE (ours) |
|---|---|---|---|---|---|---|
| LlaMa3 8B | 7.44 | 7.91 | 8.21 | 7.88 | 9.08 | 7.50 |
| Mistral 7B | 2.19 | 3.11 | 4.97 | 5.45 | 10.10 | 2.11 |
| Qwen2.5 32B | 11.20 | 11.90 | 11.8 | 12.8 | 12.30 | 11.09 |
| Gemma2 9B | 3.73 | 4.19 | 4.82 | 3.98 | 8.40 | 3.93 |
| OLMOE 7B | 8.33 | 8.75 | 10.70 | 14.60 | 11.04 | 8.55 |
| Orca2 7B | 3.97 | 4.33 | 4.95 | 7.09 | 8.20 | 3.67 |

## G   INFERENCE TIME COMPARISON

As mentioned in Section 2.2, KALE is a post-training method designed to enhance the knowledge manipulation capabilities of LLMs. **Once the model completes training, KALE maintains identical autoregressive inference characteristics to vanilla LLMs during the decoding phase, introducing zero additional temporal overhead and requiring no retrieval operations from external knowledge bases.** We conduct comparative measurements of average inference latency per sample across different methodologies (vanilla LLM, CoT, TOG, StructGPT, GraphRAG, and KALE) using an Nvidia A100 GPU (80GB). The quantitative results in Table 10 reveal that KALE achieves nearly identical inference speed to vanilla LLMs. At the inference stage, both KALE and Vanilla models follow a similar logic: they directly take the instruction and question as input to the LLM. **Therefore, any observed speed differences between them are primarily attributable to slight variations in the length of their generated outputs.** There are instances where the Vanilla model's output length is marginally longer than KALE's, leading to KALE being slightly faster, and vice versa. This minor difference in token generation directly impacts the overall inference time. In contrast, RAG-based approaches requiring knowledge retrieval and CoT methods with extended prompt sequences incur additional computational overhead.

## H   MORE RESULTS ON THE HYPERPARAMETER SENSITIVITY EVALUATION OF KALE

Regarding the sensitivity of KALE to the hyperparameters of each component, we conduct experiments to demonstrate its robustness. For all datasets, our default setting involved randomly sampling 10 anchor entities from their 3-hop neighbors. The consistent superior performance of KALE across diverse datasets under these unified parameter settings highlights its general effectiveness.

Moreover, to further investigate KALE's robustness, we conduct experiments by varying these key hyperparameters. As shown in Table 11, we can observe that KALE exhibits robustness to changes in both the number of anchor entities and hops for neighbors. This further underscores the practical potential and reliability of our KALE framework in real-world applications.

Table 11: Hyperparameter sensitivity evaluation on the number of anchor entities and the hop of neighbors.

| Anchors | Hops | Absr | ARC-c | ARC-e | Common | MMLU | BBH | RACE-h | RACE-m |
|---|---|---|---|---|---|---|---|---|---|
| 5 | 2 | 82.94 | 80.03 | 84.18 | 66.34 | 61.79 | **57.98** | 68.27 | 73.33 |
| 5 | 4 | 84.50 | 78.50 | 84.13 | 63.64 | 61.33 | 57.82 | 66.60 | 71.73 |
| 15 | 2 | 82.94 | 75.09 | 85.00 | 64.95 | 60.03 | 56.13 | 65.75 | 69.64 |
| 15 | 4 | **85.31** | 76.19 | **87.40** | 65.44 | 60.68 | 54.45 | 65.41 | 71.03 |
| 10 (ori) | 3(ori) | 83.62 | **81.23** | 86.45 | **65.69** | **63.27** | 57.33 | **68.61** | **74.12** |

# I  AVERAGE STEPS OF EXTRACTED REASONING PATHS

By default, we generate three-hop reasoning paths from questions to answers for each question-answer pair. If a 3-hop reasoning path cannot reach the answer entity, we still provide these paths as auxiliary information to facilitate rationale generation by the LLM. We currently provide the proportion of each hop within the generated

Table 12: Statistics of average step in reasoning path on the AbsR dataset.

| 1-hop | 2-hop | 3-hop complete | 3-hop partial |
|-------|-------|----------------|---------------|
| 15.76 | 54.03 | 28.27 | 1.94 |

reasoning paths, where '3hop complete' indicates that the three-hop reasoning path successfully reached the answer, and '3hop partial' indicates that the reasoning path did not reach the answer entity. As shown in Table 12, we find that most of the reasoning paths can directly lead to the final answer entity. Specifically, less than 2% of the reasoning paths cannot reach the answer entity. This suggests that the extracted reasoning paths can effectively elucidate the underlying logic and correlations between the question and the answer.

# J  PROMPT TEMPLATES

We list the prompt templates for different tasks to offer more visually intuitive results for each task. More detailed prompt information for the best performance of each task and dataset can be seen within the code.

The placeholders Known Fact, Question, Answer, Reasoning Path, Options, Generic Fact, and Rationales will be filled with the corresponding terms in each example of corresponding benchmarks.

## J.1  PROMPT TEMPLATES FOR KNOWN FACT CHECKING

> **Prompt Templates for Known Fact Checking**
>
> ```
> You are a cautious assistant.  You carefully follow
> instructions.  You are helpful and harmless and you follow
> ethical guidelines and promote positive behavior.  Question:
> Please determine whether the following statement is correct.
> You only answer 'yes' or 'no'.  Known Fact.
> ```

## J.2  PROMPT TEMPLATES FOR RATIONALE GENERATION

> **Prompt Templates for Rationale Generation**
>
> ```
> You are a cautious assistant.  You carefully follow
> instructions.  You are helpful and harmless and you follow
> ethical guidelines and promote positive behavior.  You are
> given the question:  Question.  The corresponding answer is:
> Answer.  The reasoning paths are:  Reasoning Path.  Please
> provide a detailed explanatory rationale that references
> these reasoning paths.  If you determine that the reasoning
> path is irrelevant to the current QA pair, you may generate
> rationales based on your own knowledge.
> ```

## J.3 PROMPT TEMPLATES FOR MAIN RESULTS

> **Prompt Templates for Main Results**
>
> ```
> You are a cautious assistant.  You carefully follow
> instructions.  You are helpful and harmless and you follow
> ethical guidelines and promote positive behavior.  You are
> given a question together with a few options, you should give
> an explanation first and then answer the question.  Your
> response should follow the format like Explanation: ___
> Answer: ___ Below is the Question and Options: Question
> Options
> ```

## J.4 PROMPT TEMPLATES FOR REASONING TRACE QUALITY EVALUATION

> **Prompt Templates for reasoning trace quality evaluation**
>
> ```
> You are a cautious assistant.  You carefully follow
> instructions.  You are helpful and harmless and you follow
> ethical guidelines and promote positive behavior.  You
> are given a rationale for a question.  Evaluate the given
> rationale along five dimensions--Factual Accuracy, Logical
> Validity, Coherence, Completeness, and Interpretability.
> For each dimension, output True if the rationale is
> correct or meets the criterion; otherwise, output False.
> You should produce a five-element list in the form like
> [True,True,True,True,True].  Below are the Question Question
> and the Rationales Rationales.
> ```

## K EXAMPLES OF GENERATED REASONING PATHS AND RATIONALES

We present extracted reasoning paths alongside the generated rationales for some samples to provide a more intuitive and straightforward understanding of KALE. We select one sample each from the domains of **Science, Medicine, Common Knowledge, Computer Science, Economics, and Art**. For each sample, we provided the extracted reasoning paths and the generated rationales to support a more comprehensive understanding of KALE.

> **Science Domain Example**
>
> ```
> Question:  what is the true color of the Sun?
> Answer Choices:
> A) Red
> B) Yellow
> C) White
> D) Blue
> Extracted Reasoning Paths:
> the Sun-emits->full spectrum light-integrates_into->white
> light
> Generated Rationales:
> The Sun emits light that contains the entire visible
> spectrum.  When these different colors of light are combined,
> they create white light.
> ```

---

**Medicine Domain Example**

**Question:**  Which of the following is a typical symptom of
cancer?
**Answer Choices:**
A) Weight gain
B) Persistent fever
C) Sore muscles
D) Acne
**Extracted Reasoning Paths:**
cancer-may cause->decreased resistance-may cause->persistent
fever
cancer-may cause->decreased immune function-may
cause->persistent fever
**Generated Rationales:**
The common symptoms that cancer may cause include decreased
resistance, which can lead to fever of unknown origin,
usually manifested as persistent fever.

---

**Common Knowledge Domain Example**

**Question:**  What do people use to absorb extra ink from a
fountain pen?
**Answer Choices:**
A) shirt pocket
B) calligrapher's hand
C) desk drawer
D) blotter
**Extracted Reasoning Paths:**
extra ink-absorbed by->absorbent paper-also is->blotter
fountain pen-produces->excess ink- absorbed by->blotter
**Generated Rationales:**
A blotter is a piece of special absorbent paper.  People use
it to press against fresh ink from a fountain pen to soak
up any excess, which prevents smudging and helps the ink dry
faster.

---

**Computer Science Domain Example**

**Question:**  Which protocol secures data for websites (padlock
in the address bar)?
**Answer Choices:**
A) HTTP
B) FTP
C) HTTPS
D) SMTP
**Extracted Reasoning Paths:**
websites-secure transport->TLS-implemented as->HTTPS
websites-handle->sensitive data-requires->encryption-provided
by->HTTPS
**Generated Rationales:**
Modern websites handle sensitive user data that requires
encryption to prevent interception.  HTTPS (Hypertext
Transfer Protocol Secure) is the solution; it is essentially
the HTTP protocol layered on top of a secure encryption
protocol, TLS (Transport Layer Security).

---

**Economics Domain Example**

**Question:** If demand increases while supply remains constant, what happens to the equilibrium price?
**Answer Choices:**
A) Lower equilibrium price
B) Stays the same
C) Higher equilibrium price
D) Becomes zero
**Extracted Reasoning Paths:**
demand-shifts right->demand curve-causes->higher equilibrium price
**Generated Rationales:**
A rightward shift in the demand curve, with supply held constant, leads to a higher equilibrium price. This occurs because at the original price, a shortage is created, causing buyers to compete and bid the price upward to a new equilibrium.

---

**Art Domain Example**

**Question:** The technique of dramatic light-dark contrast in painting is called:
**Answer Choices:**
A) Impasto
B) Fresco
C) Chiaroscuro
D) Sfumato
**Extracted Reasoning Paths:**
painting-contrast of->light and dark-technique named->chiaroscuro
painting-modeling of form->using dramatic light-a key feature of->chiaroscuro
**Generated Rationales:**
Chiaroscuro is the technique in painting that uses strong, dramatic contrasts between light and dark. Artists employ this method not only to create a sense of volume for modeling three-dimensional subjects, but also to produce a powerful, theatrical mood.

---

## L  MORE RESULTS OF DIFFERENT BACKBONE MODELS

As mentioned in Section 5.2, we select LlaMA3 8B, Mistral 7B, and Qwen2.5 32B as representative models in Table 1. In this section, to further demonstrate the generalization and versatility of KALE, we also conducted experiments on several popular open-source LLMs, including Gemma2 9B, OLMOE-1B-7B, and Orca2 7B. As shown in Table 13, we can still observe that our KALE method significantly outperforms existing baselines on these backbone models as well. This further demonstrates the effectiveness of our KALE approach. We also present radar charts for each backbone model to provide a more intuitive performance comparison in Figures 9 and 10. The effectiveness of our KALE across various popular open-source models further demonstrates its strong versatility and generalization capabilities.

## M  MORE RESULTS OF APPLYING DIFFERENT KGS TO EXTRACT RATIONALES

In the main experiments, we used Wikidata as the default KG for extracting reasoning paths. To further evaluate the robustness of KALE under different, smaller-scale KGs, we additionally extracted

Table 13: More results of our KALE using Gemma2 9B, OLMOE 7B, and Orca2 7B as backbone models. We **bold** the best results and underline the suboptimal results for each backbone model.

| Backbone | Category | Method | AbsR | ARC-c | ARC-e | Common | MMLU | BBH | RACE-h | RACE-m |
|---|---|---|---|---|---|---|---|---|---|---|
| Gemma2 9B | Prompt-based | Vanilla | 52.49 | 79.95 | 88.89 | 57.66 | 53.56 | 48.93 | 73.13 | 78.62 |
| | | CoT | 67.54 | 81.06 | 86.91 | 61.43 | 57.35 | 53.37 | 71.07 | 79.32 |
| | Retrieval-based | TOG | 72.04 | 79.27 | 81.65 | 63.06 | 59.31 | 51.53 | 75.99 | 79.53 |
| | | StructGPT | 59.24 | 83.28 | 86.87 | 59.30 | 61.40 | 57.98 | 79.87 | 81.27 |
| | | GraphRAG | 64.57 | 85.07 | 84.13 | 65.00 | 62.53 | 60.89 | 76.80 | 81.69 |
| | SFT-based | SFT | 61.37 | 81.06 | 89.06 | 58.97 | 55.26 | 51.38 | 74.93 | 80.78 |
| | | SDFT | 75.83 | 82.42 | 90.91 | 60.85 | 57.67 | 55.37 | 74.19 | 81.20 |
| | | DMT | 77.13 | 81.83 | 91.12 | 62.00 | 56.69 | 53.22 | 76.99 | 83.01 |
| | | MeanLearn | 72.04 | 80.20 | 89.90 | 63.06 | 58.98 | 57.36 | 75.53 | 81.20 |
| | | KG-SFT | 74.76 | 80.20 | 88.26 | 64.54 | 59.37 | 55.06 | 77.16 | 82.73 |
| | Augmented-based | STaR | 76.66 | 77.22 | 84.43 | 60.20 | 54.54 | 56.60 | 75.53 | 81.55 |
| | | AugGPT | 59.60 | 82.34 | 81.86 | 53.73 | 55.26 | 58.89 | 78.88 | 83.33 |
| | | GPT3Mix | 59.72 | 75.43 | 88.22 | 64.29 | 61.01 | 55.83 | 79.07 | 83.98 |
| | | **KALE (ours)** | **81.52** | **88.57** | **94.70** | **68.63** | **65.32** | **65.49** | **83.30** | **87.74** |
| OLMOE 7B | Prompt-based | Vanilla | 49.88 | 62.03 | 65.99 | 44.06 | 38.73 | 35.73 | 57.18 | 65.74 |
| | | CoT | 51.06 | 63.13 | 67.34 | 45.62 | 39.91 | 36.66 | 59.46 | 64.83 |
| | Retrieval-based | TOG | 54.50 | 64.42 | 69.82 | 47.26 | 40.89 | 38.34 | 60.35 | 67.75 |
| | | StructGPT | 56.87 | 65.70 | 71.12 | 51.60 | 41.61 | 40.64 | 60.03 | 69.63 |
| | | GraphRAG | 57.82 | 60.75 | 71.25 | 50.61 | 43.50 | 41.56 | 60.66 | 68.04 |
| | SFT-based | SFT | 53.31 | 63.91 | 68.52 | 49.14 | 40.43 | 37.58 | 59.18 | 69.63 |
| | | SDFT | 59.95 | 65.52 | 70.16 | 50.61 | 42.78 | 38.65 | 59.06 | 67.84 |
| | | DMT | 60.43 | 66.04 | 70.83 | 51.26 | 42.36 | 39.57 | 61.18 | 68.45 |
| | | MeanLearn | 71.09 | 66.30 | 67.80 | 54.55 | 44.21 | 43.10 | 60.03 | 72.42 |
| | | KG-SFT | 61.26 | 66.41 | 70.58 | 52.66 | 43.17 | 38.04 | 61.09 | 65.25 |
| | Augmented-based | STaR | 59.24 | 66.12 | 71.04 | 50.36 | 43.76 | 41.41 | 62.84 | 66.04 |
| | | AugGPT | 61.73 | 66.55 | 71.54 | 52.00 | 43.76 | 43.40 | 60.98 | 70.19 |
| | | GPT3Mix | 62.20 | 67.06 | 72.60 | 53.23 | 43.50 | 42.02 | 60.26 | 75.48 |
| | | **KALE (ours)** | **81.99** | **72.78** | **74.60** | **58.25** | **46.96** | **45.88** | **64.35** | **75.84** |
| Orca2 7B | Prompt-based | Vanilla | 61.37 | 68.34 | 70.75 | 47.67 | 44.09 | 37.27 | 72.36 | 75.49 |
| | | CoT | 67.77 | 70.90 | 77.40 | 50.86 | 43.77 | 39.20 | 72.58 | 75.84 |
| | Retrieval-based | TOG | 59.60 | 73.89 | 75.72 | 62.24 | 51.14 | 42.94 | 73.41 | 74.09 |
| | | StructGPT | 65.17 | 67.66 | 77.95 | 53.40 | 45.40 | 46.01 | 76.01 | 78.41 |
| | | GraphRAG | 67.06 | 69.97 | 78.87 | 54.71 | 50.75 | 47.70 | 76.02 | 75.77 |
| | SFT-based | SFT | 63.98 | 71.33 | 76.56 | 48.24 | 52.90 | 47.70 | 73.33 | 76.88 |
| | | SDFT | 76.66 | 72.53 | 75.72 | 52.33 | 52.25 | 46.63 | 73.99 | 75.14 |
| | | DMT | 75.24 | 73.55 | 77.15 | 51.27 | 52.63 | 48.31 | 73.41 | 77.30 |
| | | MeanLearn | 77.01 | 77.22 | 86.57 | 66.50 | 53.04 | 35.58 | 73.36 | 78.76 |
| | | KG-SFT | 78.91 | 72.44 | 78.87 | 52.42 | 54.00 | 48.93 | 74.01 | 76.90 |
| | Augmented-based | STaR | 71.68 | 75.00 | 81.57 | 64.53 | 45.85 | 44.33 | 75.33 | 77.30 |
| | | AugGPT | 61.73 | 73.89 | 80.05 | 53.89 | 47.81 | 44.32 | 75.24 | 78.55 |
| | | GPT3Mix | 69.79 | 74.58 | 79.67 | 54.46 | 50.75 | 45.25 | 75.53 | 77.51 |
| | | **KALE (ours)** | **83.41** | **78.16** | **88.51** | **69.62** | **61.20** | **50.77** | **78.62** | **80.02** |

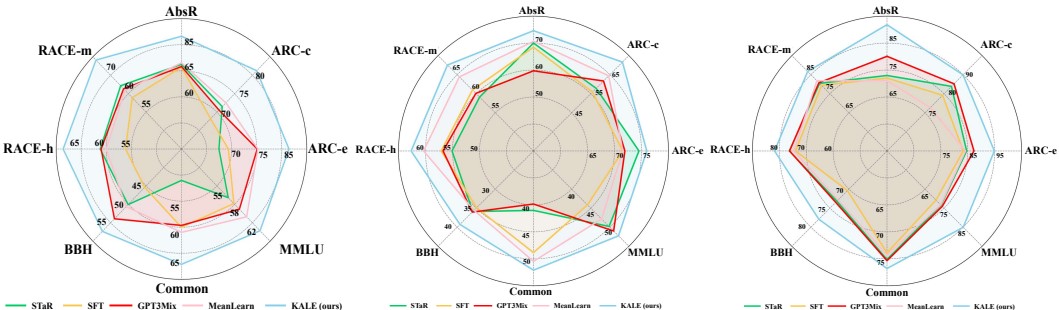

Figure 9: KALE achieves state-of-the-art performance on a broad range of scientific optimization tasks compared with existing methods, using LlaMA3 8B, Mistral 7B, and Qwen2.5 32B as backbone models, respectively.

reasoning paths from alternative KGs and generated corresponding rationales. Specifically, we employed DBpedia (Auer et al., 2007) and ConceptNet (Speer et al., 2017) to extract reasoning paths, based on which we generated rationales for training. We used LLaMA3-8B as the backbone model.

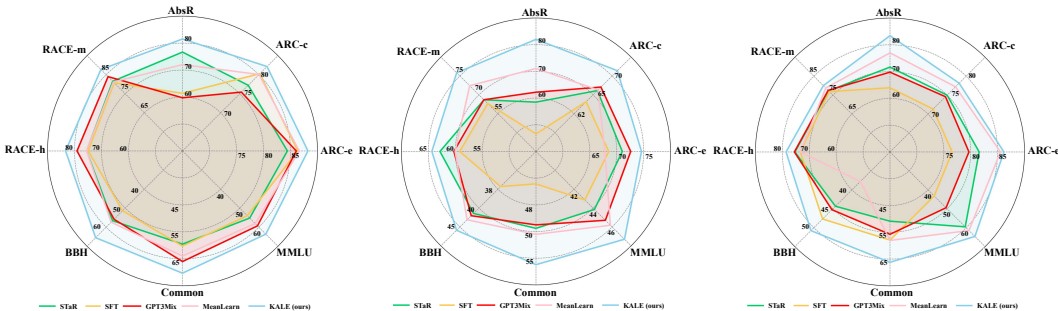

Figure 10: KALE achieves state-of-the-art performance on a broad range of scientific optimization tasks compared with existing methods, using Gemma2 9B, OLMOE 7B, and Orca2 7B as backbone models, respectively.

Table 14: Results comparison of KALE using different KGs to extract reasoning path using Llama3 8B as the backbone model.

|  | AbsR | ARC-c | ARC-e | Common | MMLU | BBH | RACE-h | RACE-m |
|---|---|---|---|---|---|---|---|---|
| $\textbf{KALE}_{DBpedia}$ | 80.81 | 77.89 | 83.77 | 60.07 | 61.28 | **58.00** | 65.58 | 68.73 |
| $\textbf{KALE}_{ConceptNet}$ | 79.93 | **81.54** | 84.19 | 62.03 | 61.06 | 55.94 | 66.93 | 71.17 |
| $\textbf{KALE}_{Wikidata}$ | **83.62** | 81.23 | **86.45** | **65.69** | **63.27** | 57.33 | **68.61** | **74.21** |

Table 15: More results of the ablation study of our KALE, using Gemma2 9B, OLMOE 7B, and Orca2 7B as the backbone models.

| Backbone | Method | AbsR | ARC-c | ARC-e | Common | MMLU | BBH | RACE-h | RACE-m |
|---|---|---|---|---|---|---|---|---|---|
| **Gemma2 9B** | $\textbf{KALE}_{\text{w/o KI}}$ | $76.54_{\downarrow4.98}$ | $84.47_{\downarrow4.10}$ | $92.17_{\downarrow2.53}$ | $65.52_{\downarrow3.11}$ | $61.14_{\downarrow4.18}$ | $61.35_{\downarrow4.14}$ | $80.02_{\downarrow3.28}$ | $84.26_{\downarrow3.48}$ |
|  | $\textbf{KALE}_{\text{w/o KA}}$ | $73.22_{\downarrow8.30}$ | $78.41_{\downarrow10.16}$ | $90.32_{\downarrow4.38}$ | $66.99_{\downarrow1.64}$ | $63.42_{\downarrow1.90}$ | $60.12_{\downarrow5.37}$ | $78.70_{\downarrow4.60}$ | $82.66_{\downarrow5.08}$ |
|  | **KALE** | **81.52** | **88.57** | **94.70** | **68.63** | **65.32** | **65.49** | **83.30** | **87.74** |
| **OLMOE 7B** | $\textbf{KALE}_{\text{w/o KI}}$ | $78.91_{\downarrow3.08}$ | $69.80_{\downarrow2.98}$ | $73.23_{\downarrow1.37}$ | $56.51_{\downarrow1.74}$ | $40.89_{\downarrow6.07}$ | $43.25_{\downarrow2.63}$ | $62.92_{\downarrow1.43}$ | $70.26_{\downarrow5.58}$ |
|  | $\textbf{KALE}_{\text{w/o KA}}$ | $74.17_{\downarrow7.82}$ | $68.26_{\downarrow4.52}$ | $70.92_{\downarrow3.68}$ | $55.28_{\downarrow2.97}$ | $44.35_{\downarrow2.61}$ | $42.48_{\downarrow3.40}$ | $60.26_{\downarrow4.09}$ | $69.22_{\downarrow6.62}$ |
|  | **KALE** | **81.99** | **72.78** | **74.60** | **58.25** | **46.96** | **45.88** | **64.35** | **75.84** |
| **Orca2 7B** | $\textbf{KALE}_{\text{w/o KI}}$ | $79.68_{\downarrow3.73}$ | $76.37_{\downarrow1.79}$ | $84.18_{\downarrow4.33}$ | $67.81_{\downarrow1.81}$ | $58.59_{\downarrow2.61}$ | $48.31_{\downarrow2.46}$ | $74.96_{\downarrow3.66}$ | $77.99_{\downarrow2.03}$ |
|  | $\textbf{KALE}_{\text{w/o KA}}$ | $77.61_{\downarrow5.80}$ | $75.43_{\downarrow2.73}$ | $82.49_{\downarrow6.02}$ | $65.52_{\downarrow4.10}$ | $54.41_{\downarrow6.79}$ | $45.86_{\downarrow4.91}$ | $73.16_{\downarrow5.46}$ | $75.91_{\downarrow4.11}$ |
|  | **KALE** | **83.41** | **78.16** | **88.51** | **69.62** | **61.20** | **50.77** | **78.62** | **80.02** |

As shown in Table 14, We observe that our KALE model exhibits relatively robust performance across different KGs. This implies a strong potential for KALE to generalize to various KGs in complex real-world datasets, thereby demonstrating its significant applicability in practical scenarios.

# N    MORE RESULTS OF ABLATION STUDY

In Section 5.3, we report the results of the ablation study using LlaMA3 8B, Mistral 7B, and Qwen2.5 32B as the backbone model. In this section, we will further present the results using Gemma2 9B, OLMOE 7B, and Orca2 7B as backbone models to obtain more insights into the individual components constituting KALE across various backbone models. As illustrated in Tables 15, we still observe that the absence of each component within KALE leads to a decline in performance across diverse domains for almost all applied backbone models in all tested benchmarks, which further demonstrates that KALE organically integrates the knowledge-induced data synthesis method and knowledge-aware fine-tuning into a unified framework as well. We still observe that the absence of knowledge-aware fine-tuning ($\textbf{KALE}_{\text{w/o KA}}$) leads to a more significant decline in accuracy, which further demonstrates the importance of effectively implicit knowledge learning.

## O MORE RESULTS OF KNOWN&INCORRECT PHENOMENON ON DIFFERENT BASELINES

As mentioned in Section 5.4, models fine-tuned using vanilla SFT still exhibit a serious known-incorrect phenomenon. In this section, we provide more analysis of the known-incorrect phenomenon to include additional baselines involving the training of LLMs. As shown in Table 16, we observe that our KALE consistently achieves the best results in knowledge manipulation analysis. If the model possesses relevant knowledge, KALE exhibits the lowest *known&incorrect* rate. Specifically, on Qwen-2.5 32B, KALE demonstrates only a 1.07% *known&incorrect* rate. This further indicates that KALE effectively enhances LLMs' knowledge manipulation ability in downstream tasks.

Table 16: Experiment results on the AbsR benchmark in six LLM backbones range for the knowledge manipulation analysis. We **bold** the best results for each method.

| Category | Method | LlaMA3 8B | OLMOE 7B | Qwen2.5 32B | Gemma2 9B | Mistral 7B | Orca2 7B |
|---|---|---|---|---|---|---|---|
| *Known&Correct* | SFT | 34.95 | 39.33 | 48.34 | 41.11 | 50.12 | 47.88 |
| | SDFT | 56.28 | 47.87 | 53.00 | 58.89 | 54.03 | 55.57 |
| | DMT | 56.64 | 44.91 | 65.17 | 61.85 | 51.78 | 57.35 |
| | Meanlearn | 48.43 | 59.12 | 60.19 | 48.93 | 56.52 | 59.12 |
| | KG-SFT | 59.83 | 50.36 | 67.06 | 55.33 | 59.36 | 62.90 |
| | STaR | 48.93 | 45.97 | 58.41 | 51.09 | 56.75 | 59.60 |
| | AugGPT | 47.27 | 45.97 | 65.76 | 43.48 | 43.01 | 48.93 |
| | GPT3Mix | 54.15 | 48.34 | 62.90 | 42.30 | 43.84 | 56.52 |
| | **KALE** | **82.94** | **79.86** | **87.56** | **77.01** | **71.09** | **75.00** |
| *Known&Incorrect* | SFT | 28.43 | 44.08 | 27.49 | 35.55 | 29.50 | 35.90 |
| | SDFT | 19.87 | 12.08 | 20.73 | 16.79 | 19.79 | 21.09 |
| | DMT | 17.93 | 15.52 | 10.07 | 15.28 | 21.44 | 17.89 |
| | Meanlearn | 22.75 | 11.97 | 10.90 | 23.11 | 14.45 | 17.89 |
| | KG-SFT | 18.37 | 10.90 | 11.85 | 19.43 | 13.03 | 16.00 |
| | STaR | 21.02 | 13.27 | 14.58 | 24.76 | 13.27 | 12.08 |
| | AugGPT | 17.18 | 15.76 | 13.15 | 16.12 | 22.27 | 12.80 |
| | GPT3Mix | 14.12 | 13.86 | 17.20 | 17.42 | 15.88 | 13.27 |
| | **KALE** | **4.15** | **2.37** | **1.07** | **2.13** | **7.7** | **8.06** |

## P MORE RESULTS OF SFT AND KALE WITH VARYING RATIOS OF TRAINING DATA

In Section 5.4, we utilized LlaMA3 8B, Mistral 7B, and Qwen2.5 32B as backbone models to investigate the performance of models trained with the SFT and KALE methods on downstream tasks under varying ratios of training rationales.

In this section, we provide additional results using other LLMs as backbone models, including Gemma2 9B, OLMOE 7B, and Orca2 7B. As shown in Figure 11, we observed that KALE demonstrated superior performance on downstream tasks even when only a small proportion of rationales was used for training. Specifically, the improvement of the OLMOE model can reach over 20% on low-data scenarios. These findings highlight the effectiveness of KALE in low-resource scenarios, which also impies a great potentials of our KALE for scenarios with limited high-quality data.

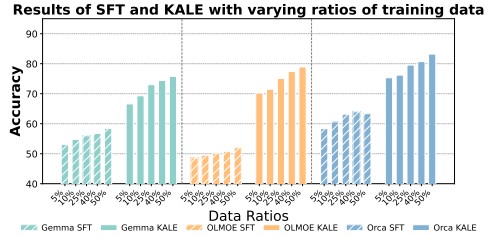

Figure 11: Results of different ratios of augmented rationales on SFT and KALE on Gemma2 9B, OLMOE 7B, and Orca2 7B, respectively.

## Q MORE RESULTS OF RATIONALES GENERATED BY DIFFERENT LLMS

In our main experiments, we utilize GPT-4.0 to generate rationales for each sample. We choose GPT-4.0 due to its exceptional performance in generating high-quality rationales, as it has demon-

Table 17: Experiment results on the AbsR benchmark in six LLM backbones range from different data ratios. We **bold** the better results for each backbone model.

| | LlaMA3 8B | | Mistral 7B | | Qwen2.5 32B | | Gemma2 9B | | OLMOE-1B-7B | | Orca2 7B | |
|---|---|---|---|---|---|---|---|---|---|---|---|---|
| % Data | SFT | KALE | SFT | KALE | SFT | KALE | SFT | KALE | SFT | KALE | SFT | KALE |
| 5% | 63.98 | **74.88** | 63.03 | **66.35** | 67.06 | **82.78** | 53.12 | **66.60** | 48.58 | **70.22** | 58.39 | **75.31** |
| 10% | 65.17 | **75.32** | 63.86 | **67.28** | 67.65 | **85.53** | 54.73 | **69.31** | 49.53 | **71.49** | 60.63 | **76.20** |
| 25% | 65.76 | **78.31** | 65.40 | **67.79** | 67.89 | **86.01** | 55.98 | **73.00** | 50.19 | **75.07** | 63.17 | **79.55** |
| 40% | 66.35 | **81.89** | 66.23 | **70.38** | 68.48 | **89.36** | 56.75 | **74.39** | 50.71 | **77.38** | 63.99 | **80.69** |
| 50% | 66.94 | **82.93** | 66.94 | **74.11** | 69.31 | **90.22** | 58.44 | **75.75** | 52.01 | **78.91** | 63.35 | **83.21** |

strated impressive results on numerous understanding and reasoning tasks. To demonstrate the generalizability of our KALE, we also incorporate two popular open-source LLMs—i.e., DeepSeek V3 and LLaMA3.1-70B-Instruct—for rationale generation and apply LLaMA3 8B as the backbone model. The results in Table 18 indicate that training on rationales generated by LLaMA3 70B and DeepSeekV3 still yields performance that significantly surpasses vanilla methods and achieves results comparable to those derived from GPT-4.0-generated rationales. **This demonstrates that KALE is relatively robust to rationales generated by different LLMs, highlighting its effectiveness for practical applications.**

Table 18: Results of KALE for rationales generated by different LLMs.

| Method | AbsR | ARC-c | ARC-e | Common | MMLU | BBH | RACE-h | RACE-m |
|---|---|---|---|---|---|---|---|---|
| Vanilla | 62.68 | 66.79 | 69.90 | 58.72 | 55.88 | 46.54 | 53.35 | 57.02 |
| KALE_DeepSeek V3 | 82.70 | **81.48** | **86.70** | 64.70 | 62.25 | **58.13** | 64.69 | 71.03 |
| KALE_Llama3 70B | 78.91 | 77.56 | 84.05 | 63.72 | 60.03 | 54.45 | 65.52 | 69.78 |
| KALE_GPT-4o (Original) | **83.62** | 81.23 | 86.45 | **65.69** | **63.27** | 57.33 | **68.61** | **74.12** |

# R MORE RESULTS OF DIFFERENT TYPES OF GENERATED RATIONALES

To further investigate whether the model genuinely benefits from meaningful knowledge or merely from the presence of any rationale, we generate two sets of modified rationales based on the original reasoning paths:

- $KALE_{irrelated}$: We instruct GPT-4o to generate factually irrelevant rationales to the reasoning paths.
- $KALE_{contrast}$: We instruct GPT-4o to generate rationales that are factually contrasting to the reasoning paths.

We denote our original method as $KALE_{ori}$ and present the comparative results in Table 19. The performance obtained using irrelevant or contrasting rationales is significantly lower than that of $KALE_{ori}$. This demonstrates the effectiveness of our knowledge-induced data synthesis, confirming that the model truly benefits from high-quality, factually accurate, and logically coherent rationales.

Table 19: Comparion of KALE on different types of generated rationales.

| | AbsR | ARC-c | ARC-e | Common | MMLU | BBH | RACE-h | RACE-m |
|---|---|---|---|---|---|---|---|---|
| $KALE_{irrelated}$ | 65.05 | 64.76 | 66.20 | 56.18 | 54.21 | 47.55 | 50.03 | 54.11 |
| $KALE_{contrast}$ | 59.60 | 59.64 | 63.56 | 51.84 | 52.25 | 42.48 | 51.11 | 48.96 |
| $\mathbf{KALE_{ori}}$ | **83.62** | **81.23** | **86.45** | **65.69** | **63.27** | **57.33** | **68.61** | **74.12** |

# S REASONING TRACE QUALITY EVALUATION OF GENERATED RATIONALES

We incorporate a reasoning trace quality metric to evaluate the quality of the generated rationales to provide more insight into our KALE. We assess rationale quality across five critical dimensions:

**Factual Accuracy, Logical Validity, Coherence, Completeness, and Interpretability** (Lee & Hockenmaier, 2025). Each dimension is evaluated as a binary classification task. Following the "LLM-as-a-judge" paradigm (Zheng et al., 2023), we utilize GPT-5 for this assessment. As shown in Table 20, we use the AbsR dataset as an example and find that rationales generated via our KALE exhibit strong performance across all five dimensions, which further validates the effectiveness of KALE.

Table 20: Reasoning trace quality evaluation for rationales on the AbsR dataset via GPT-4o.

| Factual Accuracy | Logical Validity | Coherence | Completeness | Interpretability |
|:---:|:---:|:---:|:---:|:---:|
| 98.82 | 97.63 | 99.53 | 100.00 | 99.64 |

## T  MORE RESULTS OF COMBINING SFT WITH KALE

We also conduct an additional experiment using Llama3 8B as the backbone model. We compare two approaches: our original KALE method (denoted as $KALE_{ori}$) and a sequential approach where the model is first fine-tuned with SFT and then further trained with KALE (denoted as $KALE_{joint}$)

As shown in Table 21, we find that while combining SFT first with KALE ($KALE_{joint}$) yields improvements only on some datasets. This presents a promising avenue for future work to thoroughly explore the optimal integration of KALE with existing post-training methods to achieve more consistent and significant performance enhancements for specific downstream domains.

Table 21: Comparison results of combining SFT with KALE pipeline using Llama3 8B as the backbone model.

| | AbsR | ARC-c | ARC-e | Common | MMLU | BBH | RACE-h | RACE-m |
|---|:---:|:---:|:---:|:---:|:---:|:---:|:---:|:---:|
| $KALE_{joint}$ | 80.21 | **82.25** | 84.18 | 61.34 | 60.42 | 53.22 | 67.98 | 72.91 |
| $KALE_{ori}$ | **83.62** | 81.23 | **86.45** | **65.69** | **63.27** | **57.33** | **68.61** | **74.12** |

## U  MORE DISCUSSIONS ON KALE

### U.1  WHAT NAMED ENTITY RECOGNITION METHOD IS EMPLOYED IN KALE, AND DOES IT HAVE ANY TAILORED DESIGNS?

Given the relative maturity of named entity recognition (NER), we do not elaborate on it in the main text. Considering the need for rapid deployment and ease of implementation, we utilized the SpaCy open-source library for the NER component. Moreover, we employ noun phrase extraction from the NLTK library to retain some non-named yet significant nouns in given Q&A pairs. We also reference the entity list from Wikidata for entity recognition. We also think that other specific optimized NERs are promising to improve KALE.

### U.2  WHY IS THE A* ALGORITHM EMPLOYED FOR KNOWLEDGE-INDUCED DATA SYNTHESIS INSTEAD OF THE NAÏVE BFS ALGORITHM?

In knowledge-induced data synthesis, we select the A* algorithm over the naïve BFS based on algorithmic efficiency. The A* algorithm guides the search direction by incorporating a heuristic function $h(\mathbf{n})$, which significantly reduces the exploration scope. Particularly in large-scale KGs such as Wikidata, employing BFS to identify reasoning paths often requires days of computation. As mentioned in Section 4.1, the extraction of reasoning paths from the AbsR's training set **requires over one week**. Therefore, we propose an efficient multi-path A* algorithm to extract reasoning paths. It requires **less than 4 hours** to extract all reasoning paths on the same set. Consequently, we adopt the A* algorithm as a scalable and efficient solution for reasoning path search.

### U.3    IS IT POSSIBLE FOR SOME REASONING PATHS TO NOT REACH THE ANSWER ENTITY?

During the process of extracting reasoning paths, instances may arise where the hop between the question entity and the answer entity exceeds the predefined threshold $m$. Nevertheless, the statistical data on the ABS dataset, as in Table 12, indicates that less than 2% of the 3-hop inference paths are unable to reach the answer entity. This suggests that employing 3-hop inference paths is a highly effective approach for extracting relevant information from the question to the answer. In such cases, we utilize the partial reasoning path that can be extracted-the path from the question entity to its neighboring entities within three hops—as enriched information for input. The ablation study results in Tables 2 and 15 further demonstrate the simplicity and effectiveness these types of reasoning paths.

### U.4    IF THE KG CONTAINS ERRORS THAT LEAD TO INCORRECT REASONING PATHS, WOULD GPT-4O GENERATE WRONG RATIONALES?

(i) Owing to Wikidata's factually accurate, high-quality, community-driven, and dynamically growing character, extracted reasoning paths contain **negligible** factual or logical errors. This motivates us to generate high-quality rationales via large-scale Wikidata. (ii) To address potential errors in the rationale generation, we leverage GPT's in-context learning (ICL) by incorporating specific instructions in the prompt. This allows for a filtering and correction mechanism to be implicitly applied during reasoning. As shown in Appendix J.2, we instruct LLM to generate rationales by referring to the provided reasoning path: *however, if the given reasoning path is irrelevant to the QA, generate a rationale based on your own knowledge.* This instruction helps that incorrect information is reduced. We empirically observe that utilizing the in-context learning ability is simple yet effective to reduce the error propagation with great domain robustness. A promising future work is to enable double-check mechanism with multiple state-of-the-art LLMs.

### U.5    WHAT IS THE REASON FOR CHOOSING THE KL DIVERGENCE AS THE LOSS FUNCTION?

The selection of KL divergence is due to its ability to quantify the difference between two probability distributions. It encourages the model to compress the information contained in the rationale into its parameters $\theta$. By forcing the two distributions to align, the model must "internalize" the information from the rationale $\mathbf{x}^{\text{rats}}$ into its parameters $\theta$, such that it can perform well even when $\mathbf{x}^{\text{rats}}$ is unavailable (e.g., at test time). This minimization process implicitly guides the model to capture the underlying structure of the data, thereby facilitating the learning of meaningful representations without explicit supervision. Furthermore, KL divergence is essentially composed of entropy and cross-entropy. The knowledge-aware learning module in KALE can be viewed as a distillation process, designed to enhance the knowledge manipulation capabilities of LLMs during the testing phase, where rationale input is unavailable. **The addition of KL divergence is intended to enable the model to dynamically retrieve the task-relevant knowledge it has already mastered, which improves its knowledge manipulation capability.** We also believe that a theoretical analysis of KALE, especially the KL divergence part, could lead to a deeper understanding of our KALE. We agree that this is a promising direction for future work.

## V    LLM USAGE

We used a large language model (LLM)–based writing assistant for grammar and wording improvements on draft text. The LLM did not generate research ideas, claims, proofs, algorithms, code, figures, or analyses, and it did not have access to any non-public data. During rationale generation, we use LLMs to transfer reasoning path into rationales. All edits suggested by the LLM were manually reviewed and either accepted or rewritten by the authors, who take full responsibility for the final content. The LLM is not an author of this paper.

