# OpenReview forum: "KALE: Enhancing Knowledge Manipulation in Large Language Models via Knowledge-aware Learning"
_ICLR.cc/2026/Conference — ICLR 2026 Conference Withdrawn Submission_

### Official Review · Reviewer_DHgt · 2025-10-17

**Soundness:** 4
**Presentation:** 3
**Contribution:** 3
**Rating:** 6
**Confidence:** 4

**Summary:**

The paper tackles the “known & incorrect” failure in LLMs—models hold the right facts but fail to use them—and proposes KALE, a post-training framework that strengthens knowledge manipulation (recall, reasoning, transfer).  KALE has two parts: Knowledge-Induced data synthesis that extracts multi-hop reasoning paths from external knowledge graphs and uses them to generate high-quality textual rationales, and Knowledge-Aware fine-tuning that aligns the model’s token distributions *with* and *without* these rationales by minimizing the KL divergence, encouraging the model to internalize rationale information so it can retrieve and apply relevant knowledge even when no rationale is provided at test time. Across eight benchmarks and six backbones, KALE consistently outperforms strong baselines, underscoring more reliable knowledge use.

**Strengths:**

1. This paper formalizes the “known & incorrect” gap and empirically shows this failure mode remains common after SFT, which shows a clear problem framing and strong motivation.
2. The paper uses external KGs to extract multi-hop reasoning paths → generate textual rationales (KI), then minimize KL divergence between outputs with/without rationales for knowledge-aware fine-tuning (KA), so the model can retrieve relevant knowledge even when no rationale is provided at inference. The pipeline is coherent and goal-aligned.
3. The main experiment results and ablation studies consolidate the effectiveness and scalability of the proposed frameworks.

**Weaknesses:**

1. All experiments fine-tune on each benchmark’s training set separately. This setup resembles “task-specific adaptation” rather than evaluating cross-task generalization.
2. While the authors emphasize no extra inference-time cost, training includes:
(1) path extraction from large KGs (still requires full preprocessing, though faster than BFS),
(2) GPT-4o calls for rationale generation (API cost and reproducibility issues), and
(3) KL-based consistency training (dual forward passes).
Combined, these are likely much heavier than inference-time methods.

**Questions:**

1. How does performance degrade as KG coverage/quality drops (e.g., ablate edges, introduce noise)? Any robustness to wrong or conflicting triples, and do you weight paths by confidence?
2. When no full path connects question to answer, what fraction of training pairs fall back to partial paths, and how does that affect accuracy?
3. If you inject KI-style rationales at inference (without KA training), how much do base and SFT models improve, and can base + rationales ever surpass KALE?

---

> ### Author Response · Authors · 2025-11-16
> **Response to Reviewer DHgt (Part 1 / 4)**
>
> We thank the reviewer for the insightful and valuable comments. We respond to each comment as follows and sincerely hope that our rebuttal could properly address your concerns. If so, we would deeply appreciate it if you could **raise your score ( 6: marginally above the acceptance threshold).** If not, please let us know your further concerns, and we will continue actively responding to your comments and improving our submission.
>
> > All experiments fine-tune on each benchmark's training set separately. This setup resembles “task-specific adaptation” rather than evaluating cross-task generalization.
>
> Thank you for this insightful comment. **We adopt the setting of fine-tuning on each benchmark’s training set separately, primarily to ensure consistency with prior work, such as MeanLearn [1], which follows the same protocol.** This alignment allows for a more direct and fair comparison with existing methods.
>
> **To further address the concern regarding cross-task generalization, we have conducted an additional joint training experiment.** Specifically, we fine-tune our model using the merged training sets of the eight benchmarks listed in Table 1 in the original manuscript, using Llama3 8B as the backbone model.
>
> The results from this joint training setting are summarized in Table 1. Notably, **we observe that joint training does not significantly degrade performance across the benchmarks. On the contrary, the model even achieves improved performance for the logical reasoning task on datasets including AbsR, Commonsense, and BBH.**
>
> These findings highlight that KALE is not limited to a task-specific fine-tuning paradigm. Instead, **it is compatible with multi-task or unified training setups, further demonstrating its flexibility and potential for real-world deployment scenarios** where a single model is required to handle diverse tasks.
>
> [1] Xiong, Kai, et al. "Meaningful learning: Enhancing abstract reasoning in large language models via generic fact guidance." Advances in Neural Information Processing Systems 37 (2024): 120501-120525.
>
> Table 1. Results of KALE on joint training experiments using Llama3 8B as the backbone model.
>
> | Method         | AbsR      | ARC-c     | ARC-e     | Common    | MMLU      | BBH       | RACE-h    | RACE-m    |
> | -------------- | --------- | --------- | --------- | --------- | --------- | --------- | --------- | --------- |
> | **KALE_ori**   | 83.62     | **81.23** | **86.45** | 65.69     | **63.27** | 57.33     | **68.61** | **74.12** |
> | **KALE_joint** | **84.48** | 78.16     | 84.13     | **67.81** | 60.68     | **58.13** | 64.41     | 71.57     |

---

> > ### Author Response · Authors · 2025-11-16
> > **Response to Reviewer DHgt (Part 2 / 4)**
> >
> > > While the authors emphasize no extra inference-time cost, training includes: (1) path extraction from large KGs (still requires full preprocessing, though faster than BFS), (2) GPT-4o calls for rationale generation (API cost and reproducibility issues), and (3) KL-based consistency training (dual forward passes). Combined, these are likely much heavier than inference-time methods.
> >
> > Thank you for raising this insightful point. We answer it as follows:
> >
> > ● **Performance vs. Inference-Time Methods:** Compared to inference-time methods (e.g., GraphRAG, ToG, StructGPT), KALE demonstrates significantly stronger performance. Furthermore, as shown in Table 10, the inference-time cost of KALE is comparable to that of vanilla autoregressive models, introducing no additional computational burden during inference. For your convenience, we quote Table 10 from the original manuscript as Table 2 here.
> >
> > ● **Training Cost Optimization:** While we acknowledge that KALE incurs certain training costs, we have made several optimizations to minimize these overheads:
> >
> >   **1. Efficient Knowledge Graph Reasoning Path Extraction:** Reasoning path extraction from large-scale KGs is indeed time-consuming; however, our proposed multi-path A* algorithm significantly reduces this cost. For example, extracting reasoning paths from the AbsR's training set via BFS requires over one week, whereas our method completes the same task in less than 4 hours. This optimization transforms an otherwise impractical process into a relatively efficient one, demonstrating our commitment to balancing performance and efficiency.
> >
> >   **2. Cost-Effective Model Integration:** To ensure cost control, we validated KALE using open-sourced models such as LLaMA3 and DeepSeek R1, as detailed in Table 18 in Appendix Q. These results highlight the feasibility of running the KALE pipeline within a practical budget (for your convenience, we quote Table 18 in the original manuscript as Table 3 here).
> >
> >   **3. Reproducibility:** We are fully committed to ensuring reproducibility. To this end, we will release the complete implementation, including the reasoning paths, GPT-4o-generated rationales, and training scripts, as part of our open-source package. This will enable the research community to reproduce and build upon our results with minimal effort.
> >
> > ● **KL-Based Consistency Training:** While KL-based consistency training involves dual forward passes, the implementation can be streamlined by leveraging existing frameworks such as LlamaFactory [2], significantly reducing engineering effort. This ensures that deploying KALE in practical applications remains feasible and efficient.
> >
> > [2] Zheng, Yaowei, et al. "LlamaFactory: Unified Efficient Fine-Tuning of 100+ Language Models." Proceedings of the 62nd Annual Meeting of the Association for Computational Linguistics (Volume 3: System Demonstrations). 2024.
> >
> > In summary, KALE achieves state-of-the-art performance while maintaining inference efficiency comparable to vanilla autoregressive models. At the same time, we have carefully optimized the training process to balance performance, cost, and reproducibility, making our approach both practical and impactful.
> >
> > Table 2 Average testing time for each sample on the AbsR dataset for each method (Unit: second).
> >
> > | Backbone Models | Vanilla | CoT   | TOG   | StructGPT | GraphRAG | KALE (ours) |
> > | --------------- | ------- | ----- | ----- | --------- | -------- | ----------- |
> > | LlaMa3 8B       | 7.44    | 7.91  | 8.21  | 7.88      | 9.08     | 7.50        |
> > | Mistral 7B      | 2.19    | 3.11  | 4.97  | 5.45      | 10.10    | 2.11        |
> > | Qwen2.5 32B     | 11.20   | 11.90 | 11.80 | 12.80     | 12.30    | 11.09       |
> > | Gemma2 9B       | 3.73    | 4.19  | 4.82  | 3.98      | 8.40     | 3.93        |
> > | OLMOE 7B        | 8.33    | 8.75  | 10.70 | 14.60     | 11.04    | 8.55        |
> > | Orca2 7B        | 3.97    | 4.33  | 4.95  | 7.09      | 8.20     | 3.67        |
> >
> >
> >
> > Table 3. Results of KALE for rationales generated by different LLMs.
> >
> > | Method                 | AbsR      | ARC-c     | ARC-e     | Common    | MMLU      | BBH       | RACE-h    | RACE-m    |
> > | ---------------------- | --------- | --------- | --------- | --------- | --------- | --------- | --------- | --------- |
> > | Vanilla                | 62.68     | 66.79     | 69.90     | 58.72     | 55.88     | 46.54     | 53.35     | 57.02     |
> > | KALE_DeepSeek V3       | 82.70     | **81.48** | **86.70** | 64.70     | 62.25     | **58.13** | 64.69     | 71.03     |
> > | KALE_Llama3 70B        | 78.91     | 77.56     | 84.05     | 63.72     | 60.03     | 54.45     | 65.52     | 69.78     |
> > | KALE_GPT-4o (Original) | **83.62** | 81.23     | 86.45     | **65.69** | **63.27** | 57.33     | **68.61** | **74.12** |

---

> > > ### Author Response · Authors · 2025-11-16
> > > **Response to Reviewer DHgt (Part 3 / 4)**
> > >
> > > > How does performance degrade as KG coverage/quality drops (e.g., ablate edges, introduce noise)? Any robustness to wrong or conflicting triples, and do you weight paths by confidence?
> > >
> > > We thank the reviewer for raising this important point. We answer it as follows:
> > >
> > > ● **Robustness to Missing or Noisy Knowledge:** **As shown in Table 12 of the original manuscript (quoted as Table 4 below for your convenience),** we observe that in most practical scenarios, the number of missing or erroneous KG triples is relatively low. For general-purpose reasoning tasks, Wikidata already provides sufficiently high coverage and quality to support effective path extraction.
> > >
> > > ● **Generalization Across Knowledge Graphs:** To further validate the robustness of KALE, we conducted additional experiments using alternative KGs for path extraction. **As presented in Appendix M (Table 14, quoted as Table 5 for your convenience), we found that KALE maintains strong performance across different KGs, indicating its general robustness to variations in KG structure and content.**
> > >
> > > ● **Handling Incorrect or Conflicting Triples:** For erroneous or conflicting triples, we leverage in-context learning with GPT-4o during the rationale generation phase. This design choice enables dynamic filtering or re-interpretation of conflicting information, and is intended to keep the overall system efficient and modular without adding additional complexity.
> > >
> > > ● **Confidence Weighting of Reasoning Paths:** Currently, KALE does not weight reasoning paths based on their confidence scores. However, incorporating confidence-aware path selection is indeed a promising direction for future work.
> > >
> > > Table 4 Statistics of average step in reasoning path on the AbsR dataset.
> > >
> > > | 1-hop | 2-hop | 3-hop complete | 3-hop partial |
> > > | ----- | ----- | -------------- | ------------- |
> > > | 15.76 | 54.03 | 28.27          | 1.94          |
> > >
> > > Table 5 Results comparison of KALE using different KGs to extract reasoning path using Llama3 8B as the backbone model.
> > >
> > > | Model             | AbsR      | ARC-c     | ARC-e     | Common    | MMLU      | BBH       | RACE-h    | RACE-m    |
> > > | ----------------- | --------- | --------- | --------- | --------- | --------- | --------- | --------- | --------- |
> > > | **KALE_DBpedia**| 80.81     | 77.89     | 83.77     | 60.07     | 61.28     | **58.00** | 65.58     | 68.73     |
> > > | **KALE_ConceptNet**| 79.93     | **81.54** | 84.19     | 62.03     | 61.06     | 55.94     | 66.93     | 71.17     |
> > > |**KALE_Wikidata**| **83.62** | 81.23     | **86.45** | **65.69** | **63.27** | 57.33     | **68.61** | **74.21** |

---

> > > > ### Author Response · Authors · 2025-11-16
> > > > **Response to Reviewer DHgt (Part 4 / 4)**
> > > >
> > > > > When no full path connects question to answer, what fraction of training pairs fall back to partial paths, and how does that affect accuracy?
> > > >
> > > > Thank you for these insightful comments.** As shown in Table 4 above, the proportion of training pairs for which no complete path connects the question to the answer is relatively low.** This observation highlights the effectiveness of our  KI module. **Moreover, as in Table 1 in the original manuscript, the overall performance of KALE remains stable, with minimal or no degradation, even in the presence of such cases.**
> > > >
> > > > **When a full reasoning path is unavailable, KALE leverages partial reasoning paths as auxiliary information. These partial paths often still contain semantically relevant information, contributing positively to the model's reasoning capability.**
> > > >
> > > > **For domain-specific scenarios where the KG may be sparse or insufficient to support full reasoning paths, constructing a high-quality KG can mitigate this issue.** For instance, recent advances such as SAC-KG [3] have demonstrated the ability to construct large-scale, high-quality KGs with millions of nodes directly from raw textual data. We believe that as these KG construction techniques continue to mature, KALE's applicability and effectiveness will further expand across diverse domains.
> > > >
> > > > [3] Chen, Hanzhu, et al. "SAC-KG: Exploiting Large Language Models as Skilled Automatic Constructors for Domain Knowledge Graph." Proceedings of the 62nd Annual Meeting of the Association for Computational Linguistics (Volume 1: Long Papers). 2024.
> > > >
> > > > > If you inject KI-style rationales at inference (without KA training), how much do base and SFT models improve, and can base + rationales ever surpass KALE?
> > > >
> > > > Thank you for the thoughtful question. In our implementation, **KI-style rationales leverage the ground-truth answer from the training set to serve as the end entity. However, during inference, the answer is not accessible.** To ensure a fair comparison, we can not apply KI at inference time, as it would require access to ground-truth answers or additional retrieval-based operations that are not feasible in real-world deployment.
> > > >
> > > > This design choice highlights a key advantage of our proposed method, KALE: **it does not rely on any additional retrieval or external information during inference.** KALE is fully self-contained and can be seamlessly integrated into an existing inference system (e.g., vLLM [4]) without modifying the serving pipeline. This plug-and-play capability significantly reduces engineering overhead and enhances the practicality of KALE in real-world applications.
> > > >
> > > > [4] Kwon, Woosuk, et al. "Efficient memory management for large language model serving with paged attention." Proceedings of the 29th symposium on operating systems principles. 2023.
> > > >
> > > >
> > > > ---
> > > >
> > > > **We humbly hope our response has addressed your concerns. If you have any additional concerns or comments that we may have missed in our responses, we would be most grateful for any further feedback from you to help us further enhance our work.**

---

> ### Author Response · Authors · 2025-11-26
> **We are looking forward to your further comments and/or questions.**
>
> Dear Reviewer DHgt,
>
> We deeply appreciate your valuable feedback and the time you've taken to review our work, especially during this busy period.
>
> We are reaching out to kindly inquire about the current status of your review regarding our submission. **We sincerely hope that our responses have adequately addressed your concerns. Furthermore, we are eager to address any additional queries you might have, which will enable us to enhance our work further.**
>
> Once again, thank you for your guidance and support.
>
> Best,
>
> Authors

---

### Official Review · Reviewer_aHZc · 2025-10-28

**Soundness:** 2
**Presentation:** 3
**Contribution:** 2
**Rating:** 4
**Confidence:** 4

**Summary:**

Large Language Models (LLMs) often struggle with "knowledge manipulation," failing to answer questions correctly even when they possess the necessary information, a phenomenon known as "known & incorrect." This paper proposes KALE, a post-training framework that uses Knowledge Graphs (KGs) to generate data rationales, creating structured reasoning paths for Q&A pairs.

**Strengths:**

1. The paper's problem definition is clear and significant. The "known&incorrect" phenomenon is a key pain point in LLM research. The authors clearly articulate this problem with illustrative cases, providing a strong motivation.

2. The paper proposes a novel data generation framework (KI) that combines KG paths with LLM generation. This offers a systematic method for creating high-quality reasoning data with a clear logical basis.

3. The authors conducted extensive experiments on 8 benchmarks and 6 different LLM backbones.

**Weaknesses:**

1. The attribution of efficacy for the KI synthesis stage, a core contribution in this paper, is severely confounded. The process is critically dependent on a powerful, SOTA proprietary model (GPT-4o) to "translate" KG paths into "high-quality" rationales. This makes it difficult to discern if the performance gains stem from the KALE framework's superiority or simply from distilling a stronger "teacher" model. The authors' own results in Appendix Q (Table 18) amplify this concern: using a weaker rationale generator (Llama3 70B), KALE's performance on key benchmarks (AbsR, Common, MMLU, BBH) falls below the $KALE_{w/o~KI}$ ablation baseline (Table 2). This strongly suggests KALE's success relies heavily on the external teacher's capability, not its framework's generalizability.

2. The novelty of the second core innovation—Knowledge-Aware fine-tuning—is limited. The method's use of KL divergence to align model (no rationale) and teacher (with rationale) distributions is a mature technique in knowledge and self-distillation. For instance, recent work[1] has employed nearly identical KL-divergence SFT for similar motivations. The paper lacks a sufficient comparison and differentiation from this prior art.

3. The experimental design completely omits a mainstream and powerful alternative: Outcome-Based Reinforcement Learning. This RL approach, which hypothesizes that rewarding final outcomes is sufficient for implicit reasoning, circumvents KALE's central challenge: the "lack of high-quality textual reasoning data." A discussion and empirical comparison with RL methods is strongly suggested.

[1] Efficient Knowledge Injection in LLMs via Self-Distillation.

**Questions:**

See above

---

> ### Author Response · Authors · 2025-11-16
> **Response to Reviewer aHZc (Part 1 / 4)**
>
> We thank the reviewer for the insightful and valuable comments. We respond to each comment as follows and sincerely hope that our rebuttal could properly address your concerns. If so, we would deeply appreciate it if you could **raise your score ( 4: marginally below the acceptance threshold).** If not, please let us know your further concerns, and we will continue actively responding to your comments and improving our submission.
>
> > The attribution of efficacy for the KI synthesis stage, a core contribution in this paper, is severely confounded. The process is critically dependent on a powerful, SOTA proprietary model (GPT-4o) to "translate" KG paths into "high-quality" rationales. This makes it difficult to discern if the performance gains stem from the KALE framework's superiority or simply from distilling a stronger "teacher" model. The authors' own results in Appendix Q (Table 18) amplify this concern: using a weaker rationale generator (Llama3 70B), KALE's performance on key benchmarks (AbsR, Common, MMLU, BBH) falls below the  ablation baseline (Table 2). This strongly suggests KALE's success relies heavily on the external teacher's capability, not its framework's generalizability.
>
> Thank you for your thorough review and valuable feedback. we answer it as follows:
>
> - We agree that stronger teacher models, such as GPT-4o, are indeed capable of generating higher-quality rationales, which can in turn improve the student model’s performance. However, we would like to emphasize that the **KALE is designed to extract and distill reasoning structures from these rationales—specifically via our proposed multi-path A star algorithm—rather than simply mimicking the teacher's outputs.** This distinction is critical to understanding KALE's contribution.  Indeed, as shown in Appendix Q (Table 18) **(For your convenience, we quote Table 18 as Table 1 here)**, we conduct a comparative analysis using both GPT-4o and LLaMA3-70B as rationale generators. While there is an expected performance drop when using the weaker LLaMA3-70B model, **the performance gap between KALE (GPT-4o) (63.27% on MMLU) and KALE (LLaMA3-70B) (60.03% on MMLU) is significantly smaller than the performance gap between GPT-4o (90.89% on MMLU) and LLaMA3-70B(81.56% on MMLU) themselves. This suggests that KALE narrows the performance disparity induced by teacher model quality, thereby demonstrating the robustness and generalizability of our pipeline.** Moreover, we observe that on several benchmarks (e.g., ARC-c, ARC-e, and BBH), using open-source models such as DeepSeek-R1 as the rationale generator outperforms the GPT-4o-based variant. This further indicates that KALE's success is not simply attributable to distillation from a more powerful model, but rather to the interplay between our structured KI generation mechanism and the rationale quality.
>
> -  We also highlight an ablation study in our original manuscript **(for your convenience, Tables 2 and 12 in the original manuscript; quoted below as Tables 2 and 3 here),** where we evaluate the importance of the KI-guided synthesis stage. In this ablation, we remove the A*-based KI generation and directly prompt the teacher LLM (GPT-4o) to generate rationales for each QA pair. This baseline, denoted as "KALE (w/o KI)," does not utilize our multi-path reasoning approach. As shown in Table 2, performance degrades significantly in this setting. For instance, with Mistral-7B as the backbone model, we observe a drop of 12.50% on the ARC-e dataset. **This demonstrates the importance of the KI-guided reasoning paths in generating effective rationales and supports the efficacy of our synthesis stage beyond simple distillation.**
>
> -  **We also conduct an additional experiment using LLaMA3-8B as both the teacher and student to validate the quality of our generated rationales in a self-taught setting, as shown in Table 4.** Even under this constrained setting, KALE consistently improves performance, highlighting that the reasoning paths synthesized from KI are of sufficient quality to facilitate learning even without access to a powerful teacher.
>
> In summary, while teacher model strength influences rationale quality, our results strongly support that KALE’s performance gains are not solely attributable to this factor. The multi-path reasoning process and KI synthesis strategy provide crucial, non-trivial contributions that enable consistent performance improvements across a range of settings.

---

> > ### Author Response · Authors · 2025-11-16
> > **Response to Reviewer aHZc (Part 2 / 4)**
> >
> > Table 1.  Results of KALE for rationales generated by different LLMs using Llama3 8b as the backbone model.
> >
> > | Method                   |      AbsR |     ARC-c |     ARC-e |    Common |      MMLU |       BBH |    RACE-h |    RACE-m |
> > | ------------------------ | --------: | --------: | --------: | --------: | --------: | --------: | --------: | --------: |
> > | Vanilla                  |     62.68 |     66.79 |     69.90 |     58.72 |     55.88 |     46.54 |     53.35 |     57.02 |
> > | **KALE_DeepSeek V3**     |     82.70 | **81.48** | **86.70** |     64.70 |     62.25 | **58.13** |     64.69 |     71.03 |
> > | **KALE_Llama3 70B**        |     78.91 |     77.56 |     84.05 |     63.72 |     60.03 |     54.45 |     65.52 |     69.78 |
> > | **KALE_GPT-4o (Original)** | **83.62** |     81.23 |     86.45 | **65.69** | **63.27** |     57.33 | **68.61** | **74.12** |
> >
> > Table 2 Results of the ablation study of KALE, using LlaMA3 8B, Mistral 7B, and Qwen2.5 32B as backbones.
> >
> > | Backbone        | Method            | AbsR      | ARC-c     | ARC-e     | Common    | MMLU      | BBH       | RACE-h    | RACE-m    |
> > | --------------- | ----------------- | --------- | --------- | --------- | --------- | --------- | --------- | --------- | --------- |
> > | **LlaMA3 8B**   | **KALE (w/o KI)** | 78.91     | 76.79     | 81.65     | 65.52     | 60.09     | 55.21     | 64.15     | 69.50     |
> > | **LlaMA3 8B**   | **KALE (w/o KA)** | 73.93     | 75.26     | 78.70     | 63.06     | 60.74     | 53.68     | 60.03     | 64.76     |
> > | **LlaMA3 8B**   | **KALE**          | **83.62** | **81.23** | **86.45** | **65.69** | **63.27** | **57.33** | **68.61** | **74.12** |
> > | **Mistral 7B**  | **KALE (w/o KI)** | 71.09     | 6.30      | 65.45     | 57.58     | 52.58     | 36.81     | 64.95     | 66.8      |
> > | **Mistral 7B**  | **KALE (w/o KA)** | 65.64     | 63.91     | 63.05     | 56.84     | 49.05     | 35.74     | 62.78     | 64.00     |
> > | **Mistral 7B**  | **KALE**          | **76.90** | **71.59** | **77.95** | **59.05** | **54.21** | **39.26** | **67.98** | **70.06** |
> > | **Qwen2.5 32B** | **KALE (w/o KI)** | 87.32     | 87.03     | 9.98      | 71.01     | 86.87     | 75.15     | 78.04     | 83.57     |
> > | **Qwen2.5 32B** | **KALE (w/o KA)** | 82.94     | 85.32     | 88.38     | 70.43     | 84.91     | 76.69     | 77.82     | 82.94     |
> > | **Qwen2.5 32B** | **KALE**          | **91.82** | **89.93** | **94.90** | **75.02** | **88.59** | **77.91** | **81.76** | **86.70** |
> >
> > Table 2. Results of the ablation study of KALE, using Gemma2 9B, OLMOE 7B, and Orca2 7B as the backbone models.
> >
> > | Backbone      | Method            | AbsR      | ARC-c     | ARC-e     | Common    | MMLU      | BBH       | RACE-h    | RACE-m    |
> > | ------------- | ----------------- | --------- | --------- | --------- | --------- | --------- | --------- | --------- | --------- |
> > | **Gemma2 9B** | **KALE (w/o KI)** | 76.54     | 84.47     | 92.17     | 65.52     | 61.14     | 61.35     | 80.02     | 84.26     |
> > | **Gemma2 9B** | **KALE (w/o KA)** | 73.22     | 78.41     | 90.32     | 66.99     | 63.42     | 60.12     | 78.70     | 82.66     |
> > | **Gemma2 9B** | **KALE**          | **81.52** | **88.57** | **94.70** | **68.63** | **65.32** | **65.49** | **83.30** | **87.74** |
> > | **OLMOE 7B**  | **KALE (w/o KI)** | 78.91     | 69.80     | 73.23     | 56.51     | 40.89     | 43.25     | 62.92     | 70.26     |
> > | **OLMOE 7B**  | **KALE (w/o KA)** | 74.17     | 68.26     | 70.92     | 55.28     | 44.35     | 42.48     | 60.26     | 69.22     |
> > | **OLMOE 7B**  | **KALE**          | **81.99** | **72.78** | **74.60** | **58.25** | **46.96** | **45.88** | **64.35** | **75.84** |
> > | **Orca2 7B**  | **KALE (w/o KI)** | 79.68     | 76.37     | 84.18     | 67.81     | 58.59     | 48.31     | 74.96     | 77.99     |
> > | **Orca2 7B**  | **KALE (w/o KA)** | 77.61     | 75.43     | 82.49     | 65.52     | 54.41     | 45.86     | 73.16     | 75.91     |
> > | **Orca2 7B**  | **KALE**          | **83.41** | **78.16** | **88.51** | **69.62** | **61.20** | **50.77** | **78.62** | **80.02** |
> >
> > Table 4. Results of self-taught KALE using Llama3 8B as the backbone model.
> >
> > | Method               | AbsR      | ARC-c     | ARC-e     | Common    | MMLU      | BBH       | RACE-h    | RACE-m    |
> > | -------------------- | --------- | --------- | --------- | --------- | --------- | --------- | --------- | --------- |
> > | Vanilla              | 62.68     | 66.79     | 69.90     | 58.72     | 55.88     | 46.54     | 53.35     | 57.02     |
> > | **KALE_self_taught** | **77.31** | **73.78** | **80.08** | **61.84** | **59.89** | **52.91** | **62.19** | **67.57** |

---

> ### Author Response · Authors · 2025-11-16
> **Response to Reviewer aHZc (Part 3 / 4)**
>
> > The novelty of the second core innovation—Knowledge-Aware fine-tuning—is limited. The method's use of KL divergence to align model (no rationale) and teacher (with rationale) distributions is a mature technique in knowledge and self-distillation. For instance, recent work[1] has employed nearly identical KL-divergence SFT for similar motivations. The paper lacks a sufficient comparison and differentiation from this prior art.
>
> Thank you very much for this constructive comment. We answer it in two-fold:
>
> **Motivation for KL Divergence:** Conventional cross-entropy training focuses on maximizing the likelihood of specific target outputs given specific inputs. However, our KA module is not merely to generate identical outputs regardless of the presence of rationales, but rather to encourage the model to maintain consistent distributions over answers—thereby preserving its reasoning capabilities even when rationales are absent during inference.
>
> To this end, we formulate KA based on the KL divergence between the output distributions conditioned on inputs with and without rationales. The KL divergence, defined as:
>
> $$
> KL(P∥Q)=−H(P)+H(P,Q)
> $$
>
> Where Q and P denote distributions with and without rationales as input. **It naturally contains the cross-entropy term (denoted by H(P, Q)).** It also allows us to align the output distribution conditioned on rationales with that conditioned solely on the raw input  **(denoted by H(P)).** This formulation **enables flexible distribution-level alignment rather than forcing token-level agreement during generation.**
>
> **More comprehensive Comparison of our KALE:**  Thank you for highlighting this great relevant work. We also find some similarity between this method and our KALE. **To enable a more comprehensive insight, we have now incorporated it as a baseline in our experiments, using LLaMA3-8B as the backbone.** As shown in Table 5, our full KALE pipeline consistently outperforms this prior method (referred to as prompt distillation (PD)), which we attribute to the effectiveness of our reasoning-path-based rationale generation strategy. **This further emphasizes the importance of high-quality rationales and their integration into the training pipeline.**
>
>
> Table 5. Comparison of PD and KALE using Llama3 8B as the backbone model.
>
> | Method   | AbsR      | ARC-c     | ARC-e     | Common    | MMLU      | BBH       | RACE-h    | RACE-m    |
> | -------- | --------- | --------- | --------- | --------- | --------- | --------- | --------- | --------- |
> | PD       | 75.31     | 72.17     | 76.98     | 57.73     | 60.17     | 55.41     | 58.89     | 68.91     |
> | **KALE** | **83.62** | **81.23** | **86.45** | **65.69** | **63.27** | **57.33** | **68.61** | **74.12** |

---

> > ### Author Response · Authors · 2025-11-16
> > **Response to Reviewer aHZc (Part 4 / 4)**
> >
> > > The experimental design completely omits a mainstream and powerful alternative: Outcome-Based Reinforcement Learning. This RL approach, which hypothesizes that rewarding final outcomes is sufficient for implicit reasoning, circumvents KALE's central challenge: the "lack of high-quality textual reasoning data." A discussion and empirical comparison with RL methods is strongly suggested.
> >
> > Thank you very much for this insightful and constructive comment. We answer it in two parts:
> >
> > Indeed, methods such as GRPO represent a compelling alternative by relying solely on questions and ground truth answers, thereby bypassing the need for high-quality intermediate textual reasoning annotations—the central challenge KALE is designed to address.
> >
> > To investigate this, we have conducted additional experiments using GRPO with LLaMA3-8B as the backbone, implemented via the LLaMAFactory framework. The results in Table 6 demonstrate that GRPO, when directly trained on QA samples, still underperforms compared to our proposed KALE pipeline, especially on the ARC-e dataset, which leads to 12.92%. **This empirical evidence further supports the effectiveness of KALE in enhancing the reasoning capabilities of LLMs.**
> >
> > Moreover, we note that Outcome-Based RL approaches often suffer from lower training efficiency. Specifically, RL methods require extensive sample rollouts and typically operate under sparse reward settings. This results in a high computational burden during training. In contrast, the KI component of KALE utilizes an efficient multi-path A* algorithm, while the KA component optimizes a sample-level distribution loss that provides dense supervision signals, leading to significantly more efficient training.
> >
> > **We also note that KALE and RL-based methods are not mutually exclusive.** KALE is designed to mitigate the "known and incorrect" phenomenon commonly observed in SFT models, which tend to overfit to explicit input-output mappings and fail to dynamically retrieve and manipulate task-relevant knowledge [1,2]. KALE instead enhances the model's knowledge manipulation capabilities—specifically, its ability to recall, reason, and transfer relevant knowledge effectively.
> >
> > On the other hand, RL methods often aim to elicit emergent reasoning capabilities by leveraging trial-and-error dynamics, sometimes leading to "aha moments" in reasoning. From an engineering standpoint, one could further fine-tune a KALE-pretrained checkpoint using GRPO (or similar RL techniques) to improve performance on specific downstream tasks such as long-chain-of-thought reasoning or self-correction.
> >
> > **We believe that integrating KALE into a broader LLM post-training framework that includes RL techniques is a promising direction for future research. Such a unified approach may yield deeper insights into the post-training landscape and further unlock the full potential of knowledge-aware language modeling.**
> >
> > [1] Allen-Zhu, Zeyuan, and Yuanzhi Li. "Physics of language models: part 3.1, knowledge storage and extraction." Proceedings of the 41st International Conference on Machine Learning. 2024.
> >
> > [2] Allen-Zhu, Zeyuan, and Yuanzhi Li. "Physics of Language Models: Part 3.2, Knowledge Manipulation." The Thirteenth International Conference on Learning Representations.
> >
> >
> > Table 6. Comparison of GRPO and KALE using Llama3 8B as the backbone model.
> >
> > | Method   | AbsR      | ARC-c     | ARC-e     | Common    | MMLU      | BBH       | RACE-h    | RACE-m    |
> > | -------- | --------- | --------- | --------- | --------- | --------- | --------- | --------- | --------- |
> > | GRPO     | 75.83     | 75.77     | 73.53     | 63.64     | 59.37     | 51.07     | 63.32     | 68.87     |
> > | **KALE** | **83.62** | **81.23** | **86.45** | **65.69** | **63.27** | **57.33** | **68.61** | **74.12** |
> >
> > ---
> >
> > **We humbly hope our response has addressed your concerns. If you have any additional concerns or comments that we may have missed in our responses, we would be most grateful for any further feedback from you to help us further enhance our work.**

---

> ### Author Response · Authors · 2025-11-26
> **We are looking forward to your further comments and/or questions.**
>
> Dear Reviewer aHZc,
>
> We deeply appreciate your valuable feedback and the time you've taken to review our work, especially during this busy period.
>
> We are reaching out to kindly inquire about the current status of your review regarding our submission. **We sincerely hope that our responses have adequately addressed your concerns. Furthermore, we are eager to address any additional queries you might have, which will enable us to enhance our work further.**
>
> Once again, thank you for your guidance and support.
>
> Best,
>
> Authors

---

### Official Review · Reviewer_EeXb · 2025-11-01

**Soundness:** 2
**Presentation:** 2
**Contribution:** 2
**Rating:** 4
**Confidence:** 3

**Summary:**

This paper proposes KALE, a two-stage framework to improve knowledge manipulation in large language models.
- First, a Knowledge-Induced (KI) data generation step uses external knowledge graphs (e.g., Wikidata) to extract multi-hop reasoning paths and generate rationales via GPT-4o.
- Second, a Knowledge-Aware (KA) learning paradigm minimizes the KL divergence between distributions of models trained with and without rationales, encouraging internalization of explicit reasoning.
- Experiments on multiple knowledge-intensive benchmarks (MMLU, RACE, ARC, BBH, AbsR) show consistent accuracy improvements.

**Strengths:**

1. The method elegantly integrates external structured knowledge with rationale-based learning, addressing the “known-but-incorrect” problem in LLMs.

2. The proposed model consistently improves across model scales (7B–32B), showing stable generality.

**Weaknesses:**

1. No evaluation on open-ended generation or general abilities. The paper focuses solely on accuracy in knowledge tasks and does not verify whether KALE harms general fluency or creativity after fine-tuning.

2. No comparison with modern reasoning or “thinking-style” models.
Baselines (ToG, StructGPT, GraphRAG) are early models and do not include current SOTA models like DeepSeek-R1, Qwen2.5-Think, or Llama3 thinking model. Hence, the claimed “SOTA” results may be overstated.

**Questions:**

Please see above.

---

> ### Author Response · Authors · 2025-11-16
> **Response to Reviewer EeXb (Part 1 / 2)**
>
> We thank the reviewer for the insightful and valuable comments. We respond to each comment as follows and sincerely hope that our rebuttal could properly address your concerns.If so, we would deeply appreciate it if you could **raise your score ( 4: marginally below the acceptance threshold).** If not, please let us know your further concerns, and we will continue actively responding to your comments and improving our submission.
>
> > No evaluation on open-ended generation or general abilities. The paper focuses solely on accuracy in knowledge tasks and does not verify whether KALE harms general fluency or creativity after fine-tuning.
>
> Thank you for raising this important point. Our primary focus in this work is to improve LLM performance on knowledge manipulation tasks, which are crucial for enabling models to reason and respond accurately based on existing factual and procedural knowledge, such as in mathematics and multi-hop QA tasks [1]. Therefore, our core evaluations have concentrated on QA-style benchmarks that reflect such capabilities.
>
> **We also agree that it is important to ensure that KALE does not degrade the model's general abilities, especially in open-ended generation, which requires fluency, coherence, and creativity.** To address this, we conducted an additional evaluation on open-ended generation using **MT-Bench** [2], a benchmark suite that covers a wide range of tasks, including creative and free-form responses. One such example prompt is:
>
> ```Compose an engaging travel blog post about a recent trip to Hawaii, highlighting cultural experiences and must-see attractions.```
>
> **We use KALE, SFT, and vanilla models as representative models using Llama3 8B as the backbones** and **employed GPT-5 LLM-Judge** for evaluation.
>
> When evaluating with MT-Bench, we conduct pairwise comparisons between the KALE model and both the vanilla model and the SFT model individually. To avoid Position Bias [1], the evaluation is as follows:
>
> For the same (question, Model A, Model B) instance:
>
> ● We first evaluate using the order (A, B).
>
> ● Then, we swap the responses and evaluate again using the order (B, A).
>
> Only when both judgments agree on the same winner do we count it as a "win" for that model.
> If the two judgments yield conflicting results, we record the outcome as a "tie."
>
> We report the specific proportions of win rate, loss rate, and tie rate as follows.
>
> The results in Table 1 show that the **KALE-trained model consistently outperforms both the vanilla model and the SFT models in terms of open-ended generation quality, with a significantly higher win rate from the LLM judge.** This suggests that KALE not only preserves but potentially enhances the model's general generation abilities. These findings provide evidence that **KALE does not harm, and can even improve, general fluency and creativity.** We thank the suggestion, which helped us strengthen the paper by including this broader evaluation.
>
> [1]Allen-Zhu, Zeyuan, and Yuanzhi Li. "Physics of Language Models: Part 3.2, Knowledge Manipulation." The Thirteenth International Conference on Learning Representations.
>
> [2] Zheng, Lianmin, et al. "Judging llm-as-a-judge with mt-bench and chatbot arena." Advances in neural information processing systems 36 (2023): 46595-46623.
>
> Table 1.  Comparison of vanilla,  SFT, and KALE model on open-ended generation settings.
>
> |      | Vanilla      |               |              | SFT          |               |              |
> | ---- | ------------ | ------------- | ------------ | ------------ | ------------- | ------------ |
> |      | **Win_rate** | **Loss_rate** | **Tie_rate** | **Win_rate** | **Loss_rate** | **Tie_rate** |
> | KALE | 82.5         | 6.25          | 11.25        | 77.5         | 8.75          | 13.75        |

---

> > ### Author Response · Authors · 2025-11-16
> > **Response to Reviewer EeXb (Part 2 / 2)**
> >
> > > No comparison with modern reasoning or “thinking-style” models. Baselines (ToG, StructGPT, GraphRAG) are early models and do not include current SOTA models like DeepSeek-R1, Qwen2.5-Think, or Llama3 thinking model. Hence, the claimed “SOTA” results may be overstated.
> >
> > Thank you for pointing out this issue. We answer it as follows:
> >
> > **Baseline Selection Justification:**  Our choice of LLM backbones was guided by two key considerations:
> >
> > ● **Popularity and community adoption:** These models exhibit relatively high popularity and usage in the open-source community, as reflected by their download counts on Hugging Face.
> >
> > ● **Comparability with prior work:** To ensure fair and consistent benchmarking, we aligned our baseline selection with prior works [3,4] for a fair comparison.
> >
> > [3] Xiong, Kai, et al. "Meaningful learning: Enhancing abstract reasoning in large language models via generic fact guidance." Advances in Neural Information Processing Systems 37 (2024): 120501-120525.
> >
> > [4] Chen, Hanzhu, et al. "Knowledge graph finetuning enhances knowledge manipulation in large language models." The Thirteenth International Conference on Learning Representations. 2025.
> >
> > **Comparison with More Recent “Thinking-style” Models:** Due to resource limitations, we were unable to fine-tune the full DeepSeek-R1 model. Furthermore, we do not find officially "Qwen2.5-Think" or "LLaMA3 thinking" variants in official HuggingFace model hubs. Therefore, **we conduct additional experiments evaluating our KALE pipeline on Qwen3-32B (thinking): a state-of-the-art model with enhanced multi-step reasoning capabilities.** The updated experiments show that the **KALE continues to provide significant improvements over vanilla and SFT methods when applied to these newer LLMs.** This supports our claims regarding the robustness and generalizability of our KALE across different model backbones and reasoning capabilities.
> >
> > Table 2 Results of KALE on Qwen3-32B-thinking model.
> >
> > | Method   | AbsR      | ARC-c     | ARC-e     | Common    | MMLU      | BBH       | RACE-h    | RACE-m    |
> > | -------- | --------- | --------- | --------- | --------- | --------- | --------- | --------- | --------- |
> > | Vanilla  | 82.33     | 81.06     | 84.60     | 71.99     | 83.61     | 87.38     | 77.47     | 81.48     |
> > | SFT      | 85.82     | 84.22     | 87.54     | 74.53     | 85.56     | 88.73     | 80.05     | 84.57     |
> > | **KALE** | **93.60** | **91.09** | **93.43** | **79.08** | **89.94** | **92.02** | **84.61** | **88.54** |
> >
> >
> >
> > ---
> >
> > **We humbly hope our response has addressed your concerns. If you have any additional concerns or comments that we may have missed in our responses, we would be most grateful for any further feedback from you to help us further enhance our work.**

---

> ### Author Response · Authors · 2025-11-26
> **We are looking forward to your further comments and/or questions.**
>
> Dear Reviewer EeXb,
>
> We deeply appreciate your valuable feedback and the time you've taken to review our work, especially during this busy period.
>
> We are reaching out to kindly inquire about the current status of your review regarding our submission. **We sincerely hope that our responses have adequately addressed your concerns. Furthermore, we are eager to address any additional queries you might have, which will enable us to enhance our work further.**
>
> Once again, thank you for your guidance and support.
>
> Best,
>
> Authors

---

### Official Review · Reviewer_QCEj · 2025-11-02

**Soundness:** 2
**Presentation:** 3
**Contribution:** 3
**Rating:** 4
**Confidence:** 4

**Summary:**

This paper proposes knowledge-aware learning (KALE), which consists of two key components: knowledge-induced data synthesis (KI) to generate high-quality relational data, and knowledge-aware fine-tuning (KA) to enable large language models (LLMs) to better manipulate task-relevant knowledge. For KI, the method finds multiple reasoning paths that connect a question entity to an answer entity using the A* algorithm, where a heuristic function is derived from anchor entities. These rationales are expected to provide high-quality textual reasoning data that bridges the question and answer. For KA, the approach minimizes the divergence between the generative distributions with and without the KI-based rationales. Experimental results on various benchmarks demonstrate that KALE outperforms several baseline models, validating its effectiveness.

**Strengths:**

1.	The proposed knowledge-aware learning (KALE) framework, which integrates knowledge-induced data synthesis (KI) and knowledge-aware fine-tuning (KA), is novel and interesting. The rationale extraction process is well designed to enhance efficiency through the use of anchor entities and a three-step BFS strategy.
2.	The experimental results demonstrate that the proposed KALE framework achieves notable performance improvements, and the ablation studies clearly illustrate the individual effects of KI and KA.
3.	The presentation is clear, well-structured, and generally easy to follow, with good overall organization.

**Weaknesses:**

1.	The extracted rationales are regarded as a form of Chain-of-Thought (CoT), but it remains unclear why KA is formulated based on the KL divergence between the generative distributions with and without rationales. What is the motivation for using KL divergence, compared to more conventional fine-tuning approaches that jointly generate both rationales and answers under an autoregressive loss?
2.	From a data augmentation perspective, it is unclear why the generated dataset is relatively large compared to those of other baseline methods. How does the size of the automatically augmented dataset compare quantitatively to other approaches?
3.	The proposed rationale extraction relies on named entity recognition (NER) and a graph-based search using the A* algorithm. However, GPT-generated rationales could also be used for training on question–answer pairs. Why does the model exhibit improved performance on test questions for which such rationales are not available? It also remains unclear how the model performs when rationales for question entities are absent in the training data.

**Questions:**

1.	It remains less convincing that the proposed rationale extraction process is truly necessary. Could simpler or alternative search-based methods address this problem as effectively?

---

> ### Author Response · Authors · 2025-11-16
> **Response to Reviewer QCEj (Part 1 / 3)**
>
> We thank the reviewer for the insightful and valuable comments. We respond to each comment as follows and sincerely hope that our rebuttal could properly address your concerns. If so, we would deeply appreciate it if you could raise your score **(4: marginally below the acceptance threshold).** If not, please let us know your further concerns, and we will continue actively responding to your comments and improving our submission.
>
> > The extracted rationales are regarded as a form of Chain-of-Thought (CoT), but it remains unclear why KA is formulated based on the KL divergence between the generative distributions with and without rationales. What is the motivation for using KL divergence, compared to more conventional fine-tuning approaches that jointly generate both rationales and answers under an autoregressive loss?
>
> Thank you very much for this constructive comment.  We answer it in two-folds:
>
> - **Motivation for KL Divergence:** Conventional cross-entropy training focuses on maximizing the likelihood of specific target outputs given specific inputs. However, our KA module is not merely to generate identical outputs regardless of the presence of rationales, but rather to encourage the model to maintain consistent distributions over answers—thereby preserving its reasoning capabilities even when rationales are absent during inference.
>
> To this end, we formulate KA based on the KL divergence between the output distributions conditioned on inputs with and without rationales. The KL divergence, defined as:
>
> $$ KL(P∥Q)=−H(P)+H(P,Q) $$
>
> Where Q and P denote distributions with and without rationales as input. **It naturally contains the cross entropy term (denoted by H(P,Q))**. It also allows us to align the output distribution conditioned on rationales with that conditioned solely on the raw input  **(denoted by H(P))**. This formulation **enables flexible distribution-level alignment rather than forcing token-level agreement during generation.**
>
> - **Empirical Comparison with Cross-Entropy-Based Fine-Tuning**: We also provide an ablation study in the main paper (in Section Ablation on KL divergence) to empirically validate the necessity of KL divergence for effective knowledge alignment. **For your convenience, we quote Tables 2 and 15 in the original manuscript as Tables 1 and 2 here.** As shown in the following table, attempting to align LLMs' outputs with and without rationales using cross-entropy (denoted by KALE w/o KA) does not achieve satisfactory results. For example, on the ARC-e dataset using Mistral-7B, replacing the KL-based KA loss with a cross-entropy results in a 14.90% drop in accuracy. This demonstrates that simply minimizing the cross-entropy between outputs with and without rationales is significantly less effective.
>
> These findings confirm that KL divergence provides a more principled and effective mechanism for aligning model behavior in the presence and absence of rationales. Thus, it is a core component of our proposed loss function and essential to the success of our KALE framework. We also believe that a theoretical analysis of KALE, especially the KL divergence part, could lead to a deeper understanding of our KALE. We leave this as a promising direction for future work.
>
> Table 1 Results of the ablation study of KALE, using LlaMA3 8B, Mistral 7B, and Qwen2.5 32B as backbones.
>
> | Backbone        | Method            | AbsR      | ARC-c     | ARC-e     | Common    | MMLU      | BBH       | RACE-h    | RACE-m    |
> | --------------- | ----------------- | --------- | --------- | --------- | --------- | --------- | --------- | --------- | --------- |
> | **LlaMA3 8B**   | **KALE (w/o KI)** | 78.91     | 76.79     | 81.65     | 65.52     | 60.09     | 55.21     | 64.15     | 69.50     |
> | **LlaMA3 8B**   | **KALE (w/o KA)** | 73.93     | 75.26     | 78.70     | 63.06     | 60.74     | 53.68     | 60.03     | 64.76     |
> | **LlaMA3 8B**   | **KALE**          | **83.62** | **81.23** | **86.45** | **65.69** | **63.27** | **57.33** | **68.61** | **74.12** |
> | **Mistral 7B**  | **KALE (w/o KI)** | 71.09     | 6.30      | 65.45     | 57.58     | 52.58     | 36.81     | 64.95     | 66.8      |
> | **Mistral 7B**  | **KALE (w/o KA)** | 65.64     | 63.91     | 63.05     | 56.84     | 49.05     | 35.74     | 62.78     | 64.00     |
> | **Mistral 7B**  | **KALE**          | **76.90** | **71.59** | **77.95** | **59.05** | **54.21** | **39.26** | **67.98** | **70.06** |
> | **Qwen2.5 32B** | **KALE (w/o KI)** | 87.32     | 87.03     | 9.98      | 71.01     | 86.87     | 75.15     | 78.04     | 83.57     |
> | **Qwen2.5 32B** | **KALE (w/o KA)** | 82.94     | 85.32     | 88.38     | 70.43     | 84.91     | 76.69     | 77.82     | 82.94     |
> | **Qwen2.5 32B** | **KALE**          | **91.82** | **89.93** | **94.90** | **75.02** | **88.59** | **77.91** | **81.76** | **86.70** |

---

> ### Author Response · Authors · 2025-11-16
> **Response to Reviewer QCEj (Part 2 / 3)**
>
> Table 2. Results of the ablation study of KALE, using Gemma2 9B, OLMOE 7B, and Orca2 7B as the backbone models.
>
> | Backbone      | Method            | AbsR      | ARC-c     | ARC-e     | Common    | MMLU      | BBH       | RACE-h    | RACE-m    |
> | ------------- | ----------------- | --------- | --------- | --------- | --------- | --------- | --------- | --------- | --------- |
> | **Gemma2 9B** | **KALE (w/o KI)** | 76.54     | 84.47     | 92.17     | 65.52     | 61.14     | 61.35     | 80.02     | 84.26     |
> | **Gemma2 9B** | **KALE (w/o KA)** | 73.22     | 78.41     | 90.32     | 66.99     | 63.42     | 60.12     | 78.70     | 82.66     |
> | **Gemma2 9B** | **KALE**          | **81.52** | **88.57** | **94.70** | **68.63** | **65.32** | **65.49** | **83.30** | **87.74** |
> | **OLMOE 7B**  | **KALE (w/o KI)** | 78.91     | 69.80     | 73.23     | 56.51     | 40.89     | 43.25     | 62.92     | 70.26     |
> | **OLMOE 7B**  | **KALE (w/o KA)** | 74.17     | 68.26     | 70.92     | 55.28     | 44.35     | 42.48     | 60.26     | 69.22     |
> | **OLMOE 7B**  | **KALE**          | **81.99** | **72.78** | **74.60** | **58.25** | **46.96** | **45.88** | **64.35** | **75.84** |
> | **Orca2 7B**  | **KALE (w/o KI)** | 79.68     | 76.37     | 84.18     | 67.81     | 58.59     | 48.31     | 74.96     | 77.99     |
> | **Orca2 7B**  | **KALE (w/o KA)** | 77.61     | 75.43     | 82.49     | 65.52     | 54.41     | 45.86     | 73.16     | 75.91     |
> | **Orca2 7B**  | **KALE**          | **83.41** | **78.16** | **88.51** | **69.62** | **61.20** | **50.77** | **78.62** | **80.02** |
>
> > From a data augmentation perspective, it is unclear why the generated dataset is relatively large compared to those of other baseline methods. How does the size of the automatically augmented dataset compare quantitatively to other approaches?
>
> Thank you for your valuable feedback and for pointing out this important clarification.We apologize for any confusion regarding the size of our augmented dataset in comparison to those of other baseline methods. In fact, our automatically augmented dataset is identical to the original dataset or those used by other baselines. We clarify this below.
>
> Our proposed KI method performs instance-level augmentation, in which for each input sample, we extract reasoning paths using the proposed Multi-path A* algorithm. We then leverage GPT-4o to generate rationales based on these extracted paths. This process ensures that each original sample corresponds to one augmented instance with enhanced rationales.
>
> In the rare cases where the reasoning path extraction fails to meet the desired quality **(denoted by 3-hop partial with only 1.94% and we provide a statistical analysis of such cases in Table 12 of Appendix I, and for your convenience, we quote the corresponding results from the original manuscript in Table 3 below)**, we fall back on in-context learning (ICL). Specifically, we use a carefully designed GPT-4o prompt that instructs the model to generate rationales **using its own reasoning capability** when the provided path is unavailable or irrelevant. **For your convenience we quote the corresponding template in Appendix J from the original manuscript here as below**
>
> ```You are a cautious assistant. You carefully follow instructions. You are helpful and harmless, and you follow ethical guidelines and promote positive behavior. You are given the question: {Question}. The corresponding answer is: {Answer}. The reasoning paths are: {Reasoning Path}. Please provide a detailed explanatory rationale that references these reasoning paths. If you determine that the reasoning path is irrelevant to the current QA pair, you may generate rationales based on your own knowledge.```
>
> **Therefore, the number of augmented samples generated by our KI remains identical to the number of samples in the original dataset, and is not larger compared to other baselines.**
>
> Table 3：Statistics of average step in reasoning path on the AbsR dataset.
>
> | 1-hop | 2-hop | 3-hop complete | 3-hop partial |
> | ----- | ----- | -------------- | ------------- |
> | 15.76 | 54.03 | 28.27          | 1.94          |

---

> ### Author Response · Authors · 2025-11-16
> **Response to Reviewer QCEj (Part 3 / 3)**
>
> >The proposed rationale extraction relies on named entity recognition (NER) and a graph-based search using the A* algorithm. However, GPT-generated rationales could also be used for training on question–answer pairs. Why does the model exhibit improved performance on test questions for which such rationales are not available? It also remains unclear how the model performs when rationales for question entities are absent in the training data.
>
> Thank you for your thorough review and valuable feedback. We answer it as follows:
>
> - **Why Does the Model Improve Without Rationales at Test Time:**  Our proposed KA method utilizes rationales only during training by distilling the behavior induced by rationale-augmented inputs into models that do **not require rationales at inference time. This process is akin to knowledge distillation, where the model learns to emulate the internal reasoning patterns induced by rationales, without depending on them at test time.** Specifically, we use the rationale-enhanced model distribution as a soft target and train the model with an additional KL divergence loss to align the rationale-free predictions during test time. This approach allows the model to **internalize the reasoning path** during training and **retrieve task-relevant knowledge** during inference, even in the absence of explicit rationales.
>
> - **How Does the Model Handle Missing Rationales in Training:**  We acknowledge that there are some rare cases where rationales for question entities are not present in the training data, as shown in the statistical summary in Table 3 above.
>
> In the rare instances where reasoning paths cannot be constructed, KALE will **utilise the ICL ability of GPT-4o to generate rationales on the fly** from the raw question–answer pairs. Therefore, there exists a rationale for each sample in our pipeline.
>
> > It remains less convincing that the proposed rationale extraction process is truly necessary. Could simpler or alternative search-based methods address this problem as effectively?
>
> Thank you for your insightful comment. We address this in two parts:
>
> - **Effectiveness of Reasoning Path-Based Rationale Generation:**  Our ablation study in the original manuscript (quoted as Tables 1 and 2 above) **evaluates the impact of generating rationales withoutleveraging the KI-generated reasoning path.** Specifically, we prompted the LLM to generate rationales conditioned solely on the question and answer pair, omitting the intermediate reasoning steps. As shown in the results, the performance of such rationales was consistently inferior, indicating that reasoning path-based rationale generation provides essential contextual scaffolding. These findings support the necessity and effectiveness of our proposed KI.
>
> - **Comparison with Search-Based Rationale Generation:** To further explore alternative methods, we implemented a search-based rationale generation baseline using the BM25 [1] algorithm. Entities extracted from the question and answer were used as search queries, and the retrieved passages were employed as rationales for knowledge augmentation (denoted as search-based rationales). As shown in Table 4, while this method offers a reasonable baseline, **it underperforms compared to our KI-generated rationales, especially on the AbsR dataset, where the performance dropped by 11.26%.** This substantial gap highlights the limitations of purely search-based rationales and demonstrates the superior effectiveness of our approach in generating contextually relevant and task-adapted rationales.
>
> [1] Jones, K. S., Walker, S., & Robertson, S. E. (2000). A probabilistic model of information retrieval: development and comparative experiments: Part 2. Information processing & management, 36(6), 809-840.
>
>
> Table 4 Comparison results between search based rationales and original KI generated rationales.
>
> | Method               | AbsR      | ARC-c     | ARC-e     | Common    | MMLU      | BBH       | RACE-h    | RACE-m    |
> | -------------------- | --------- | --------- | --------- | --------- | --------- | --------- | --------- | --------- |
> | **KALE_seach_based** | 72.36     | 75.55     | 74.51     | 60.44     | 58.07     | 51.69     | 60.89     | 65.32     |
> | **KALE_ori**         | **83.62** | **81.23** | **86.45** | **65.69** | **63.27** | **57.33** | **68.61** | **74.12** |
>
> ---
>
> **We humbly hope our response has addressed your concerns. If you have any additional concerns or comments that we may have missed in our responses, we would be most grateful for any further feedback from you to help us further enhance our work.**

---

> ### Author Response · Authors · 2025-11-26
> **We are looking forward to your further comments and/or questions.**
>
> Dear Reviewer QCEj,
>
> We deeply appreciate your valuable feedback and the time you've taken to review our work, especially during this busy period.
>
> We are reaching out to kindly inquire about the current status of your review regarding our submission. **We sincerely hope that our responses have adequately addressed your concerns. Furthermore, we are eager to address any additional queries you might have, which will enable us to enhance our work further.**
>
> Once again, thank you for your guidance and support.
>
> Best,
>
> Authors

---

### Note · Authors · 2026-01-05

I have read and agree with the venue's withdrawal policy on behalf of myself and my co-authors.